

# Identification of the cloud base height over the central Himalayan region: Intercomparison of Ceilometer and Doppler Lidar

K.K. Shukla[1,2], K. Niranjan Kumar[3], D.V. Phanikumar[1], Rob K Newsom[4], V. R. Kotamarthi[5]
Taha B.M.J. Ouarda[3,6] and M. Venkat Ratnam[7]

[1]Aryabhatta Research Institute of observational sciences, Nainital, India
[2]Pt. Ravishankar Shukla University, Raipur, Chhattisgarh, India
[3]Institute Center for Water and Environment, Masdar Institute of Science and Technology, Abu Dhabi, United Arab Emirates
[4]Pacific Northwest National Laboratory, Richland, Washington, USA
[5]Argonne National Laboratory, Argonne, Illinois, USA
[6]INRS-ETE, Quebec City (Qc), Canada
[7]National Atmospheric Research Laboratory, Gadanki, Tirupati, India

*Correspondence to*: D V Phanikumar (phani@aries.res.in; astrophani@gmail.com)

**Abstract.** We present the measurement of cloud base height (CBH) derived from the Doppler Lidar (DL), Ceilometer (CM) and Moderate Resolution Imaging Spectroradiometer (MODIS) satellite over a high altitude station in the central Himalayan region for the first time. We analyzed six cases of cloud overpass during the daytime convection period by using the cloud images captured by total sky imager. The occurrence of thick clouds (> 50%) over the site is more frequent than thin clouds (< 40 %). In every case, the CBH indicates less than 1.2 km, above ground level (AGL) observed by both DL and CM instruments. The presence of low level clouds in the height-time variation of signal to noise ratio of DL and backscatter of CM shows a similar diurnal pattern on all days. Cloud fraction is found to be maximum during the convective period. The CBH estimated by the DL and CM showed reasonably good correlation ($R^2$=0.76). The DL observed updraft fraction and cloud base vertical velocity also shows good correlation ($R^2$=0.66). The inter-comparison between DL and CM will have implications in filling the gap of CBH measurements by the DL, in absence of CM. More deployments of such instruments will be invaluable for the validations of meteorological models over the observationally sparse Indian regions.

*Key words*: Cloud base height, Doppler Lidar, Celiometer, Radiosonde



## 1. Introduction

The Earth's shortwave and longwave radiation at the surface and as well as the top of the atmosphere is influenced by cloud microphysical properties such as cloud coverage and cloud base height (CBH) (Considine et al., 1997; Meerkötter and Bugliaro, 2009). The formation of all weather clouds occurs in lowest layer of the atmosphere (i.e. troposphere). The extensive occurrence of stratocumulus and stratus clouds over ocean (~34%) and land surface (~18%) in the lower atmosphere and near the atmospheric boundary layer (ABL) is well documented (Heymsfield, 1993; Considine et al., 1997). It was found that there is an increase in planetary albedo and a decrease in shortwave radiation at the surface due to ABL clouds (Heymsfield, 1993; Berg and Kassianov, 2007). Moreover, clouds can also affect the structure of atmospheric parameters like ABL height, temperature and relative humidity because of their vital role in altering the water cycle over the Earth's surface and play a critical role in the removal of atmospheric pollutants through precipitation (Ghate et al., 2011).

A strong coupling is observed between the fair weather ABL cumulus clouds and associated turbulence in the ABL, which have impact on the ABL diurnal variability (Brown et al., 2002). These clouds can be lifted more than a few hundred meters due to the ABL evolution during morning to the afternoon hours over the land (Meerkötter and Bugliaro, 2009). The cloud top height can be retrieved with different retrieval algorithms (Forsythe et al., 2000; Hutchison, 2002; Huang et al., 2006; Weisz et al., 2007) for use with various satellite observations such as the Cloud-Aerosol Lidar and Infrared Pathfinder Satellite Observations (CALIPSO) (Winker et al., 2003), CloudSat and Tropical Rainfall Measuring Mission (TRMM) (Stephens et al., 2002; Kummerow et al., 1998).

Due to the various feedbacks between clouds, radiation and dynamics described above, it is extremely important to have the simultaneous observations of the fair weather cumulus clouds and vertical velocity in the Earth's atmosphere for the appropriate representation in the Global Circulation Models (GCMS) (Randall et al., 1985; Tonttila et al., 2011). The vertical structure of convective and cumulonimbus clouds are studied by using precipitation radar over the south Asian region (Bhat and Kumar, 2015). Sharma et al., (2016) studied the CBH observed by using Ceilometer (CM) during 2013-2015 over the western site in India and also compared with the Moderate Resolution Imaging Spectroradiometer (MODIS) satellite. The observed CBH by ground-based (ceilometer) and space-based satellite (MODIS) observations are showing good correlation over the western Indian site.

In addition, observations of vertical velocity remain sparse over most of the site in the Indian region and in particular over regions with the high altitude and complex topography. The atmospheric radiation measurements (ARM) Mobile Facility (AMF1) conducted a field campaign during June 2011-March 2012 over a high altitude site Manora Peak (29.4° N; 79.2° E; 1958 m amsl), Nainital to have a better understanding about the cloud, precipitation and aerosols in the Ganges basin, i.e. Ganges Valley Aerosol eXperiment (GVAX). During GVAX, different ground based remote sensing instruments were operated to measure the atmospheric dynamical parameters. A Doppler Lidar (DL) was continuously operated to measure the vertical velocity and backscatter from the fair-weather ABL clouds. Along with the DL, CM and Total Sky Imager (TSI) were also operated continuously to measure CBH and to capture the cloud images, respectively, in the daytime over the observational site.





In the current study, our main objective is to evaluate the capability of DL in estimating the CBH and
comparing the results with CM, the standard instrument for the CBH measurement and CBH derived from the
MODIS. The advantage of DL over the other two measurements is that one can get the simultaneous information on
both vertical velocities near clouds along with the CBH. In this study, we have considered six selected cases based
on cloud coverage, residence time of clouds and availability of simultaneous datasets with other ground based
instruments over the site during the observational period (05-10 UT).
**2. Observational site, instrumentation and methodology**
Ganges valley region of the Indian subcontinent is a heavily populated region and shows an increase in the
pollutants level in the current climate (Ramanathan et al. 2005; Lau and Kim 2006; Bollasina et al., 2011). Bollasina
et al., (2011) showed the increment in the concentration of anthropogenic aerosols over the Ganges valley region
which has the ability to modify the Indian Summer Monsoon (ISM) rainfall. To understand climate change over the
Indo-Gangetic Plain (IGP), the geographical location of Manora Peak ($29.4^o$ N; $79.2^o$ E; 1958 m amsl), Nainital is
suitable for measuring the various atmospheric parameters in campaign mode. To have a better understanding of the
impact of measured parameters like aerosol, convection, cloud, and radiative characteristics of the Indian monsoon,
Atmospheric radiation measurement (ARM) mobile facility conducted a field campaign over the site which is
known as Ganges Valley Aerosol Experiment (GVAX) (Kotamarthi, 2010).  The GVAX campaign was utilized to
quantify the impact of aerosols on ISM, role of atmospheric boundary layer in aerosol transportation in the Ganges
valley region and also the effect of aerosols in the cloud formation. By considering all the above serious issues, the
current observational site is selected to conduct the campaign mode observations (Kotamarthi, 2013).
The observational site Manora Peak, Nainital is located in the central Gangetic Himalayan region and is
considered as a high altitude site in the northern part of the Indian region. It is away from the urban/industrial
pollution. The total population of the Nainital is ~ 0.5 million (according to census 2011) with population density ~
50 persons per $km^2$. The small-industries having cities i.e. Haldwani and Rudrapur are located ~20-40 km away in
the south of the observational site. A mega city, New Delhi, the capital of India is located ~ 225 km in the southwest
of the study region (Sagar et al., 2015). The site is surrounded by a dense forest. The maximum and minimum
temperatures are observed to be ~ 20 $^0$C and 1 $^0$C during pre-monsoon (March April May) and winter (December
January February) seasons, respectively (Dumka et al., 2014; Shukla et al., 2014). Moreover, wind patterns over the
site during monsoon and winter are southwesterly and northwesterly, respectively. The seasonal change in wind
pattern every year persists over the Indian subcontinent (Asnani, 2005). The detailed description about the site and
current research works carried out over the observational site can be found in detail in Sagar et al. (2015).

**2.1 Total Sky Imager (TSI)**
The TSI is manufactured by Yankee Environmental Systems (YES), and is commercialized version of the
hemispheric Sky Imager prototype (Long et al., 2006). The TSI 440/660 was deployed over the Manora Peak,





Nainital during the GVAX to capture the cloud images during the daytime. The sky cloud images captured by TSI are 24-bit color JPEG images at 352x288 pixel resolution. TSI captures the cloud image at every 30 sec during daytime. In order to retrieve cloud information, we have processed the raw cloud images of the TSI. Sky cover retrieval from TSI images is valid only for solar elevation angles $>10^0$ (zenith angles $< 80^0$) and images are processed for a $160^0$ field of view, ignoring the $10^0$ of sky near the horizon. It has a sun-blocking strip mask, which represents the location of the sun with a yellow dot in the image. We have used TSI images to infer the presence of clouds over the site for a subsequent CBH estimate. The TSI observations of cloud images are also utilized for the estimation of percentage of thin and opaque clouds over the site. The detailed discussion about the estimation of cloud properties by using TSI images are given somewhere else in previous reports (Long et al., 2001, 2006; Morris, 2005).

**2.2 Doppler Lidar**

DL was operated over a high altitude site to measure the temporal and altitude resolved vertical velocity and attenuated backscatter. In order to retrieve the radial velocity by using Doppler principle, DL uses aerosols as tracer in the atmosphere to observe the Doppler shift. The influence of insects or pollen is less in the DL observations because the small aerosols in the background dominate the signal. The DL uses an eye-safe laser of wavelength ~1.5 μm. It provides the vertical velocity and attenuated backscatter at a spatial resolution of ~30m and a temporal resolution of 1 sec. The DL can scan the atmosphere in different modes (i.e. vertically Fixed-Beam Stare (FPT), Range-Height Indicator (RHI) scan and Plan-Position Indicator (PPI) scan mode). The RHI and PPI scan modes are known as the elevational and azimuthal scan of the atmosphere, respectively. A detailed technical description of the DL system can be found in previous studies (Pearson et al., 2009; Newsom, 2012; Shukla et al., 2014). In the current study, the vertically fixed-beam stare mode of the DL is used to estimate CBH. To minimize/remove random noise fluctuations in the DL data, a threshold on signal to noise ratio (SNR) of -20 dB is applied.

**2.3 Laser Ceilometer**

The Väisälä laser Ceilometer (CT25K) is deployed over the site for precise measurements of the CBH, vertical visibility and vertical profile of aerosol backscatter during GVAX (Väisälä Oyj, 2002). It has an eye-safe laser source of wavelength ~905 nm. It provides the information at a temporal resolution of 16 sec and a spatial resolution of 30 m in the atmosphere (Morris, 2012). The 16 sec interval data is aggregated to 1 min for better comparison with the DL.

**2.4 Surface Meteorology System**

The in-situ sensors are used to measure the surface temperature (T), relative humidity (RH), pressure, wind speed and wind direction by the ARM surface meteorology systems (MET). The in-situ sensors are installed at specific



standard heights for measurement of meteorological parameters (i.e. T & RH at 2 m; Barometric pressure at 1 m and
wind speed and direction at 10 m) (Ritsche and Prell, 2011). The MET sensors provide the data at a temporal
resolution of 1-min and we have averaged for 10-min from 1-min data to calculate the lifted condensation level
(LCL) for comparison with the CBH of DL and CM, respectively.

**2.5 Radiosonde**

Väisälä Radiosondes (RS-92) were launched during GVAX at 00, 06, 12 and 18 UT daily regularly. The profiles of
atmospheric parameters (temperature, relative humidity and winds) are measured by the Radiosonde (RS) at a
vertical resolution of 10 m as the ascent rate of balloon is 5 $ms^{-1}$ and transmitter time resolution is 2 sec. In the
current study, we have used the 06 UT (11.5 hr LT) data of RS to calculate the LCL for all the cloud cases. The
detailed description about the RS can be found in previous reports (Holdridge et al., 2011; Shukla et al., 2014).

**2.6 Moderate resolution imaging spectroradiometer**

In addition to the ground based remote sensing techniques used for the estimation of CBH, we have also utilized the
MODIS satellite derived CBH over the observational site. The MODIS Terra data is obtained for the same cases as
measured by the ground based remote sensing instruments. However, the spatial resolution of MODIS cloud data is
of $1^0$ x $1^0$ latitude-longitude grids. We have used the cloud top pressure, cloud optical depth and effective radius of
liquid cloud for all the cases.

**3. Retrieval of Cloud base height (CBH) and Lifting Condensation Level (LCL)**

**3.1 Cloud Statistics from the DL**

We have used the vertical velocity and cloud statistics derived data of the DL during GVAX (Newsom et al., 2015).
In addition to clear-air vertical velocity statistics, we can derive the CBH, cloud fraction, cloud base vertical velocity
and cloud base updraft fraction. For the current data set, the averaging interval was 30 min oversampled for every 10
min. The cloud fraction is the fraction of time during the averaging interval that a cloud is detected at any height.
Similarly, the cloud base updraft fraction is the fraction of time that a positive (upward) cloud base vertical velocity
is observed during the averaging interval.

32        CBH estimates are obtained from the 1-sec DL data by detecting the heights of sharp spikes in the range-

corrected SNR. In order to identify the cloud bases, the DL uses a narrow Gaussian filter in the scattered signal due
to the presence of clouds in the atmosphere. The Gaussian filter is convolved with each SNR profile to detect the
cloud base. To minimize false detections, the CBH algorithm uses a method based on the first derivative of the
range-corrected SNR. First, the range-corrected SNR profiles are differentiated using a simple central-difference
approximation. When a cloud is present in the profile, the first derivative shows a strong positive peak immediately





below and a strong negative peak immediately above the cloud base. The algorithm then locates the maximum in the range-corrected SNR between these two extrema. Additional checks are applied to minimize false detections by rejecting temporally isolated peaks. The cloud base vertical velocity is then simply the vertical velocity at the CBH. The vertical velocity and cloud statistics derived from the DL reports the median CBH and the median cloud base vertical velocity over a given 30-min averaging interval. Further details are given in Newsom et al. (2015).

**3.2 CBH retrieval by using CM**

The measurement of the CBH with CM is known as standard method of the ground-based active remote sensing technique. The time delay between the transmitted and backscattered signal from the haze, fog, virga, mist and precipitation to the receiver of CM can be used to estimate the CBH. By knowing the time delay in equation (1), CBH can be estimated as

$$\text{Cloud base height (h)} = (c*t/2) \qquad (1)$$

where $c$ (= $3 \times 10^8$ m s$^{-1}$) is the speed of light and t is the time delay. The backscattering coefficient is estimated by using the strength and attenuation of the backscattered signal from the atmosphere. Cloud base is identified by the strong increase of the backscatter coefficient and three layers of clouds can be detected if the lower clouds are transparent (Emeis et al., 2009; Morris, 2012).

**3.3 CBH Retrieval by MODIS**

The CBH from the MODIS is calculated by taking the difference between the cloud top height and the thickness of cloud ($\Delta Z$) which is given in equation (2).

$$Z_{\text{Cloud base height}} = Z_{\text{Cloud top height}} - (\Delta Z) \qquad (2)$$

where $\Delta Z$ is the cloud thickness and $\Delta Z$ is ratio of LWP and LWC.

The thickness of water cloud depends on the relation between liquid water path (LWP) and liquid water content (LWC). Liou (1992) showed that the relation of cloud optical thickness ($\tau$) and effective radius of cloud particle size ($r_{eff}$) with LWP is given by

$$\text{LWP} = (2* \tau* r_{eff})/3 \text{ g.m}^{-2} \qquad (3)$$

LWC=0.26 g.m$^{-3}$ taken for cumulus cloud in clean condition (Hess et al., 1998).

$$\Delta Z = (\text{LWP/LWC}) \qquad (4)$$

By using LWP & LWC in equation (4), we have calculated the thickness of cloud ($\Delta Z$).

We have estimated the CBH for water cloud present in the atmosphere by using equation (2) & (4). Based on the method described by Hutchison (2002) and Sharma et al., (2016), we have used the cloud top pressure and liquid water path during daytime from MODIS Terra satellite over the observational site for cloud passages observed by the TSI.





**3.4 Lifted condensation level estimation by using surface MET and RS datasets**
The estimation of water vapor content from surface MET data has been derived by using equation (5) with T and
RH of surface meteorology (Goff-Gratch, 1946).
$$e_s = e_{st} * 10^Z \qquad (5)$$
where
$$Z = A\left(\frac{T_s}{T} - 1\right) + B \times \log_{10}\left(\frac{T_s}{T}\right) - C \times \left[10^{D\left(1-\frac{T}{T_s}\right)} - 1\right] + F\left[10^{H\left(\frac{T_s}{T}-1\right)} - 1\right]$$
and A = -7.90298, B= 5.02808, C=-1.3816 X $10^{-7}$, D= 11.344, F=8.1328 X $10^{-3}$, H= -3.49149 are the constants. $e_{st}$
(=1013.246 mb) is saturation vapor pressure ($e_s$) at boiling temperature ($T_s$=373.16 K) at standard atmospheric
pressure. By using saturation vapor pressure ($e_s$)  from equation (5) and surface RH in equation (6), we have
calculated the water vapor pressure (e)
$$RH = \left(\frac{e}{e_s}\right) \times 100 \qquad (6)$$
Dew point temperature ($T_d$) estimation by using surface MET vapor pressure (e) is given by the equation (7)
$$T_d = \left(\frac{1}{\left[\left(\frac{1}{T_0}\right) - \left(\frac{R_v}{L}\right) * ln\left(\frac{e}{e_0}\right)\right]}\right) \qquad (7)$$
where    $T_0$=273 K,      $e_o$ = 0.611 kPa,      $\frac{R_v}{L} = 0.0001844\ K^{-1}$,     e - vapor pressure
By knowing the temperature (T) and dew point temperature ($T_d$) from surface meteorology and RS, we have
calculated LCL by using equation (8)
Lifting Condensation level (LCL) height (km) = 0.125* (T-$T_d$)          (8)
**4. Results and discussion**
Figure 1 shows one of the six (12 October 2011, 21 November 2011, 11 December 2011, 20 January 2012, 08
February 2012 and 14 March 2012) cloud case examples considered in this study observed by TSI for the estimation
and comparison of CBH by different instruments over the observational site. It shows the raw (Figure 1a) and
masked (Figure 1b) cloud images by TSI at hourly interval during daytime from (10.5-15.5 hr) on 12 October 2011.
The "yellow dot" in the TSI masked image represents the position of the sun, not obscured by the clouds. However,
if this "yellow dot" becomes "white" then the sun is obscured completely by the clouds (Figure 1b; Pfister et al.,
2003). It is also to be noted that the presence of cloud is clearly apparent with the raw image of the sky captured by
TSI (Figure 1a). However, the masked images strongly confirm the presence of clouds and further distinguish
between the thin and opaque clouds by the color of the image. For instance, the blue, gray, and white colors in





Figure 1b represent the cloud free-sky, thin and opaque clouds, respectively. While the black color in Figure 1b
represent the masked pixels which are not used in determining the microphysical property of cloud by the TSI.

3         Temporal variation of masked images of clouds captured by TSI for all cases in the $160^0$ field of view

(FOV) centered at zenith in the cloud images during 10.5-15.5 LT is shown in Figure 2. Due to masked sky images,
the loss of about 17 % of the hemispherical solid angle of the sky dome is resulted. In the analysis of clear/cloudy
pixels, these masked 'black' parts are ignored (Long et al., 2006). From Figure 2, it is clearly seen that there are
lesser clouds in the forenoon (before 12.5 LT) in comparison to afternoon (after 12.5 LT) on 12 October, 21
November and 11 December 2011. We have observed the clouds at every hour on 20 January, 08 February and 14
March 2012.

10        In Figure 3 (a-f), we also show the temporal variation of the percentage occurrence of thin (shown by black

line with black open circle) and opaque clouds (red line with red open rectangle) for all the six cloud overpasses
over the observational site. To classify the thin and opaque clouds, we have performed the red-green-blue (RGB)
pixel classification of the cloud images. The scattering of blue light is more than red in clear skies and no aerosols
conditions (i.e. molecular scattering). If cloud is present in the atmosphere then red pixel value is greater than where
no clouds.

16        For clouds the relative ratio of red/blue pixel is greater than clear sky. Koehler at al. (1991) developed the

cloud decision threshold (red/blue ratio) algorithms for thin and opaque clouds for the first time from the TSI. If the
relative red/blue ratio of the pixel is higher than 0.6 then it is categorized as cloudy and lesser values are marked as
cloud free. The detailed descriptions about the classification of thin and opaque clouds are given in Slater et al.,
(2001). In most of the cases of figure 3(a-f), the percentage of opaque clouds are greater than percentage of thin
clouds. The dominance of opaque clouds is clearly seen from the figure 3(a-f) during afternoon (12.5 LT) hours over
the site. It is also evident from Figure 3 that the percentage occurrence of opaque clouds is more frequent over the
site relative to the thin clouds during the observational period.

24        Figures 4 (a1-f1) and (a2-f2), illustrate the height-time variation of SNR and backscatter for different cases

of cloud passage over the observational site observed by DL and CM, respectively. Figure 4 depicts the presence of
ABL clouds over the site. The development of convective clouds in the lowest part of ABL is due to the presence of
convective thermals. These convective thermals are crucial in the formation of the clouds because these thermals can
rise from the surface to the top of the mixing layer without being diluted (Crum and Stull, 1987). It should be noted
that the presence of the convective clouds in the ABL can be confirmed by using the observed CBH from DL and
CM and lifted condensation level (LCL) estimated from the surface (Stull and Eloranta, 1985; Zhang and Klein,
2013). During the convection, the maximum SNR is observed due to the presence of low level ABL cumulus clouds.
Also, the observed cloud cases show different dynamics of the cumulus clouds over the site. Figure 4(a1) shows the
SNR maximum around 11.5-12.5 LT   showing high percentage of opaque cloud during 12.5-14.5 LT   and then
dominated by a thin clouds, consistent with Figures 4(a1) and 4(b1). Other cases also depict similar variation with
opaque clouds more frequent than the thin clouds during convection (see Figures 4b1-f1). Similarly, Figure 4 (a2-f2)
shows the height-time variation of averaged backscatter ($srad^{-1}.km^{-1}.10^{-4}$) by the CM observed for all cloud cases in





the study. It is interesting to note that the temporal evolution and duration of thin and opaque clouds in both the
instruments are in reasonable agreement during all events.

3        In Figure 5 (a-f), we have plotted the temporal variation of CBH (with DL & CM) and cloud occurrence

frequency (with DL). The detailed description about the estimation of CBH is given in the section 3.1. The
fraction of time that a cloud is detected at any altitude during the given averaging period is defined as cloud
frequency. Varikoden et al. (2011) showed that the occurrence of low level clouds are more in comparison to the
mid-level clouds by using CM over a tropical station Akkulam, Thiruvananthapuram (8.29∘ N, 76.59∘ E, 15 m
above sea level) in India. They have also showed that the occurrence of low level clouds is higher during the
afternoon hours. We have also found that the frequency of occurrence of clouds is higher during afternoon in the
observed cases with both CM and DL over a high altitude site.

11       Figure 6 depicts the temporal variation of CBH observed by the DL and CM along with lifted condensation

level (LCL) height estimated by using surface MET parameter and RS on (a) 12 October 2011 (b) 21 November
2011 (c) 11 December 2011 (d) 20 January 2012 (e) 08 February 2012 and (f) 14 March 2012. There is a strong co-
relation between the CBH observed by the DL and CM for all cases. On an average, the CBH from both the
instruments is higher during the convective period and is associated with the change in LCL in the ABL during
daytime. ABL cloud heights are estimated by using LCL (Stackpole, 1967). The well-mixed ABL air parcels which
have a dry-adiabatic temperature profile and a constant mixing ratio are used to determine the LCL profile (Craven
et al., 2002). For the detection of CBH, the LCL is a good approximation as the CBH depends on the relative
humidity and temperature near the surface. The LCL depends on the temperature and dew point temperature above
the surface and thus a good proxy for CBH. We have estimated the LCL with surface MET and RS to compare with
the CBH of DL and CM. In 12 October 2011 case, a small difference is observed between the CBH (DL) and LCL
heights but LCL heights with the MET and RS shows a similar pattern as CBH (CM) implying the strong
association with ABL dynamics (Jones et al., 2011). From Figure 6 (a-f), it is clearly observed that in all the cases,
CBH is coupled with the LCL estimated from the surface meteorological parameters. This strong dependence of
CBH with LCL suggests the link between cloud formation and development of convection on the surface (Zheng et
al., 2015; Zheng and Rosenfeld, 2015).

27       In Figure 7, we have plotted the temporal variation of CBH with cloud base vertical velocity for all cases.

CBH observed with both the instruments are showing similar temporal variation throughout the observational time
period (10.5-15.5 LT). From figure 7(a-f), it is clearly evident that the updrafts are dominant due to the diurnal
evolution of convective ABL during daytime over the site. In some cases like 12 October 2011, 21 November 2011
and 08 February 2012, the vertical velocity follows the similar pattern. We have also plotted the temporal variation
of cloud base vertical velocity with cloud base vertical velocity updraft fraction (m) for all cases in Figure 8 (a-f).
From this figure, it is clearly seen that both the parameter are well correlated.

34       We have also compared the CBH calculated by the DL and CM with the MODIS derived CBH for all cloud

passes over the observational site. For instance, Figure 9 shows the MODIS Terra derived CBH and the daily mean
(05-10 UT) CBH measured by the DL and CM. We have taken the mean of latitude/longitude ± 1 degree by
centering the latitude/longitude of the observational site. The observed CBH from MODIS is well within the





estimated standard deviation from ground based CBH. It shows reasonably good agreement with the estimation of
CBH from the ground based and DL and CM CBH in all the cases except in two cases (21 November 2011 and 14
March 2012) where the differences are slightly higher and need to be investigated for the possible inconsistencies.
Further, we have used the DL and CM CBH as well as cloud updraft and cloud base vertical velocity
observed by the DL for all six cases to see the correlation which is plotted in Figure 10. The correlation of CBH
between the DL and CM is shown in Figure 10a. It is noticed that the CBH estimated by the DL is well correlated
($R^2$=0.76) with the CM measured CBH when we combine all the cases shown in Figure 10a. In addition, Figure 10b
illustrates the relation between cloud base vertical velocity and cloud updraft fraction observed by DL for all cloud
passes over the observational site. As indicated in Figure 10b, a strong correlation is also noted between these two
parameters. Further, it is noticed that when the cloud updraft fraction is less than 40%, the cloud base vertical
velocity tend to be negative. However, positive vertical velocities are noted when the cloud updraft fraction is more
than 50%. Kollias et al. (2001) showed that the cloud base vertical velocity is consistent with the updraft speed. We
have also observed similar behavior between the cloud base vertical velocity and updraft fraction although our
observations are from a high altitude location.
Jeong and Li (2010) estimated the CBH by using micropluse Lidar for few case studies by applying the
threshold condition of aerosol particle diameter less than 1 μm and relative humidity 40 % over the southern great
plain site. They have observed the cumulus cloud on all cases and found the CBH varying in between 1-4 km, above
mean sea level (amsl). A detailed comparison of CBH estimated over various parts of the world by using different
ground based instruments and satellite datasets is shown in Table-1. Despite different site morphologies, our CBH
values observed with both DL and CM (Table-1) are in agreement with past studies across the globe. Bühl et al.,
(2015) observed the cloud and vertical velocity by using different ground based instruments e.g. DL, cloud radar and
wind profiler over meteorological observatory, Lindenberg, Germany.
The observations from all the instruments at this site with cloud layer overhead are ~3 km. The presence of
large-scale updrafts in the cloud layers was also observed. The observed vertical velocity in the cloud layer varied
between ± 1.5 ms$^{-1}$. Similar characteristics were observed at Manora Peak, Nainital. We have observed that cloud
base vertical velocity varies between ± 2 ms$^{-1}$ except for 20 January 2012 during which higher vertical velocities of
0-4 ms$^{-1}$ were obtained.  The observed CBH also varies between 2.3-2.7 km amsl in both instruments over Manora
Peak, Nainital. Hirsch et al. (2011) retrieved the CBH by CM, and observed the shallow cumulus cloud during
daytime and CBH at 1.6±0.3 km, amsl. Also, Meerkötter and Bugliaro, (2009) estimated the CBH by using
MSG/SEVIRI, NOAA satellite data and CM data for convective cloud cases over the seven test stations near
Germany and neighboring countries.  By using geostationary satellite and ground based CMs, they have observed
that CBH varies between ~ 2-3 km and also showed a significant correlation.  Thus, our results are in good
agreement with the temporal variation of CBHs observed by DL compared with CM in previous studies.

**5. Summary and Conclusions**
In this study, we have presented comparison of the CBH estimated by using the DL with CM and MODIS derived
CBH over a high altitude site in the central Himalayan region. Total sky imager shows the presence of cloud over



the site for the cases evaluated in the current study and also opaque clouds are more frequently observed than thin clouds over the site during the observational period. The height-time variation of signal to noise ratio of DL and backscatter by the CM depict a similar pattern for the cases evaluated with opaque (thin) clouds dominating during morning (afternoon) hours in most of the cases. Strong correlation ($R^2$=0.76) between DL and CM CBH is observed suggesting that DL can also be used as a potential instrument for measuring CBH apart from standard instrument CM. Similarly, we have observed the good correlation ($R^2$=0.66) between cloud base vertical velocity and cloud updraft fraction. We have observed a similar temporal variation between CBH (estimated from DL and CM) and LCL height (Surface MET and RS) during all the cases. The CBH height and LCL height derived from surface MET and RS are also comparable. The estimated CBH with the MODIS data is also in close agreement with the ground based instruments in most of the observed cases.

Further, our results also show close agreement with the CBH derived by DL, CM and MODIS derived satellite data sets in all cases. Therefore, our results depict that DL is also a potential instrument for cloud studies apart from standard CM instrument in measuring the CBH. The cooling and warming of the atmosphere is governed by the presence of clouds at different altitudes in the atmosphere (Kiehl and Trenberth, 1997). CBH of low level clouds coupled with shallow convection is playing an essential role in the parameterization of weather and climate models (Chandra et al., 2015). Also the uncertainty observed in climate models is due to low-level clouds (Bony and Dufresne, 2005) especially when model grid spacing is much larger than the size of low level. Therefore, the continuous estimation of CBH will be a useful input for the models. Further, the cloud radiative cooling, relative humidity in the ABL and cloud cover have direct association with the low altitude clouds (Brient and Bony, 2012). Therefore, the accurate and systematic measurements of low level cloud base become important for the improvement of the models.

Hence, in this report we investigated the potential of the DL in measuring the CBH over the site in comparison to CM. From the current study, it is also clearly seen that we can use DL for CBH study over the site. It also demonstrates that the precise observations of the CBH over the complex topography are very useful for model validation. By considering the importance of the current study, CBH estimations by DL along with the cloud updraft velocities will be utilized in our future studies as potential inputs for numerical weather prediction models over the Central Himalayan region.

**Acknowledgments**

This work has been carried out as a part of GVAX campaign under joint collaboration among Atmospheric Radiation Measurement (ARM), Department of Energy (US), Indian Institute of Science (IISc) and Indian Space Research Organization (ISRO), India. We thank Director, ARIES for providing the necessary support. We also thank MODIS team for providing the valuable datasets used in the present study. The authors wish also to thank the Associate Editor, for his judicious comments which helped improve the quality of the paper.



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



1    **Table-1**: Comparison of cloud base heights with other locations around the world

| Observational site (Latitude/longitude/elevation) | Instrument | Date | Cloud base height (km) (amsl) | References |
|---|---|---|---|---|
| Lindenberg, Germany. | Doppler Lidar, Cloud radar, wind profiler | 30 July 2013 | 2.9 km | Bühl et al.,2015 |
| Israel (31.89$^0$ N , 34.81$^0$ E, 60 m) | Ceilometer | 22 April 2010 | 1.6±0.3 km | Hirsch et al., 2011 |
| Southern Great Plain (36.6$^0$ N , 97.5$^0$ W) | Micro pulse Lidar | 07, 13 and 22 May 2003 | 4.2, 1.6 and 1.3 km, respectively | Jeong and Li,2010 |
| Seven test station near Germany and neighboring countries | MSG/SEVIRI NOAA Ceilometer | 23,30 May and 30 July 2007 | Between 2-3 km | Meerkötter and Bugliaro, 2009 |
| Nainital, India (29.4$^0$ N, 79.2$^0$ E, 1958 m) | Ceilometer Doppler Lidar | 12 Oct, 21 Nov, 11 Dec, 2011 20 Jan, 08 Feb, 14 Mar 2012 | 2.468  **2.328** 2.298  **2.228** 2.688  **2.568** 2.438  **2.418** 2.678  **2.658** 2.348  **2.258** | Current study |

2    * Bold values in table represent the CBH of Ceilometer


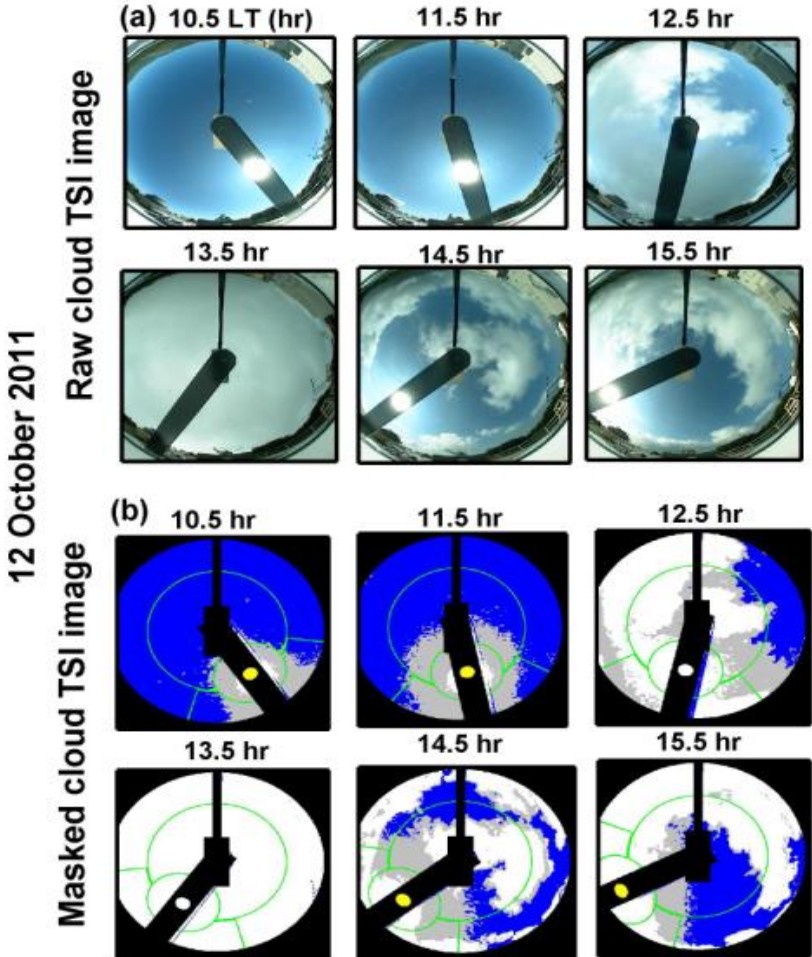

2
3
4      **Figure 1: (a-b)** Top panels show raw images of clouds during daytime observed by the TSI and bottom panels are
5          the TSI cloud decision images. In the cloud decision images, blue, gray and white colors represent cloud-free sky,
6          thin cloud and opaque clouds, respectively.  Black color represents the masked pixels which are not used in the
7          estimation of cloud property. The yellow dot on the sun-blocking strip mask represents the location of sun in the
8          image.



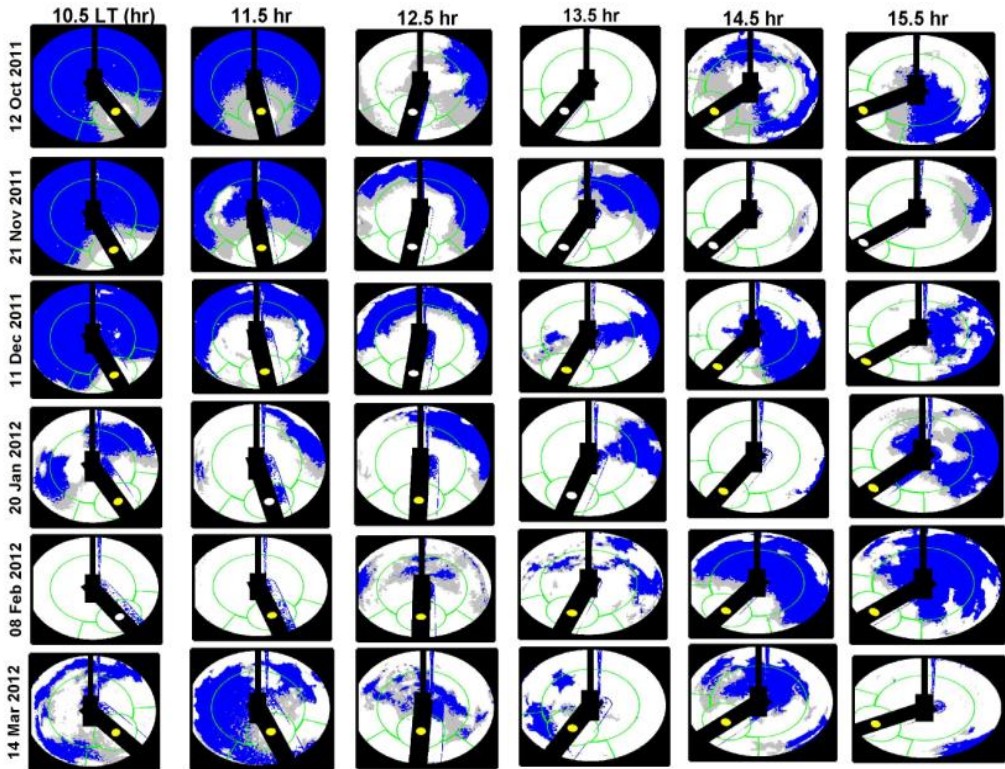

**Figure 2:** Masked images of cloud during daytime (10.5-15.5 hr) taken by the TSI for (a) 12 October 2011, (b) 21
November 2011, (c) 11 December 2011, (d) 20 January 2012, (e) 08 February 2012, and (f) 14 March 2012. In the
cloud decision images, blue, gray and white colors represent cloud-free sky, thin cloud and opaque clouds,
respectively. Black color represents the masked pixels which are not used in the estimation of cloud property. The
yellow dot on the sun-blocking strip mask represents the location of sun in the image.




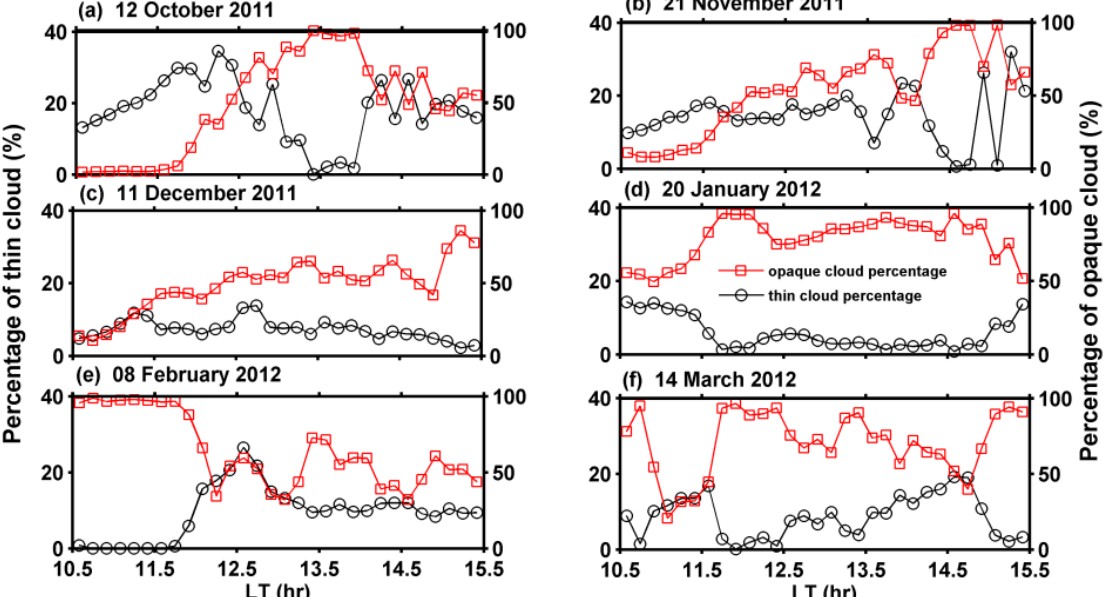

**Figure 3:** Temporal variation of (a) Percentage occurrence of thin clouds and (b) Opaque clouds observed by total sky imager over the site during (a) 12 October 2011, (b) 21 November 2011, (c) 11 December 2011, (d) 20 January 2012, (e) 08 February 2012, and (f) 14 March 2012.





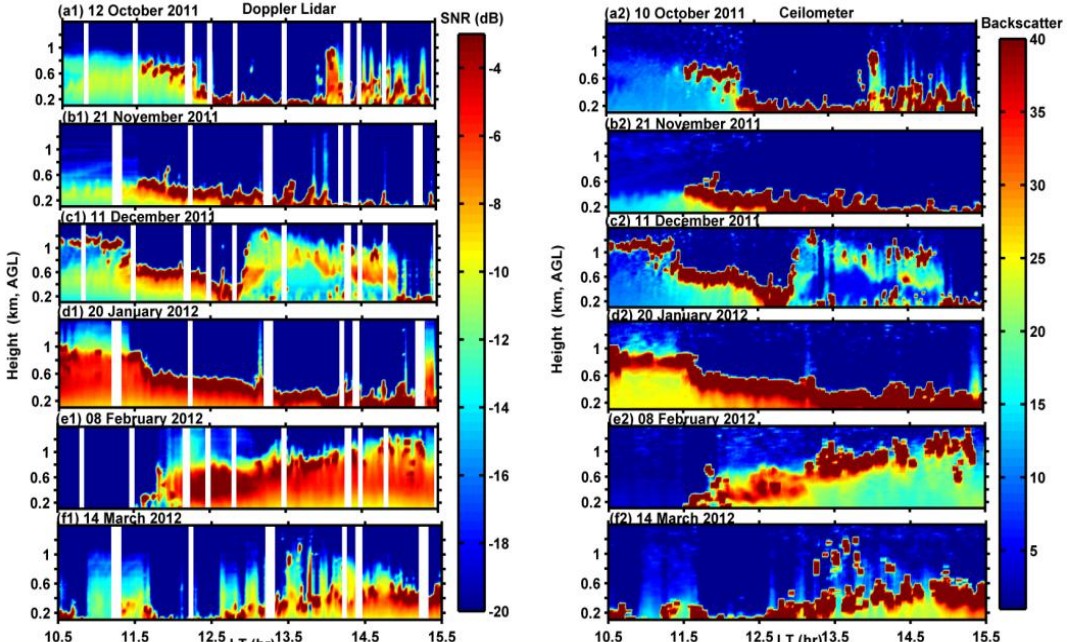

8  **Figure 4:** (a1-f1) Height-time variation of signal to noise ratio by Doppler Lidar and  (a2-f2) Height-time variation

9     of backscatter  observed by the Ceilometer during (a) 12 October 2011, (b) 21 November 2011, (c) 11 December

10     2011, (d) 20 January 2012, (e) 08 February 2012, and (f) 14 March 2012.





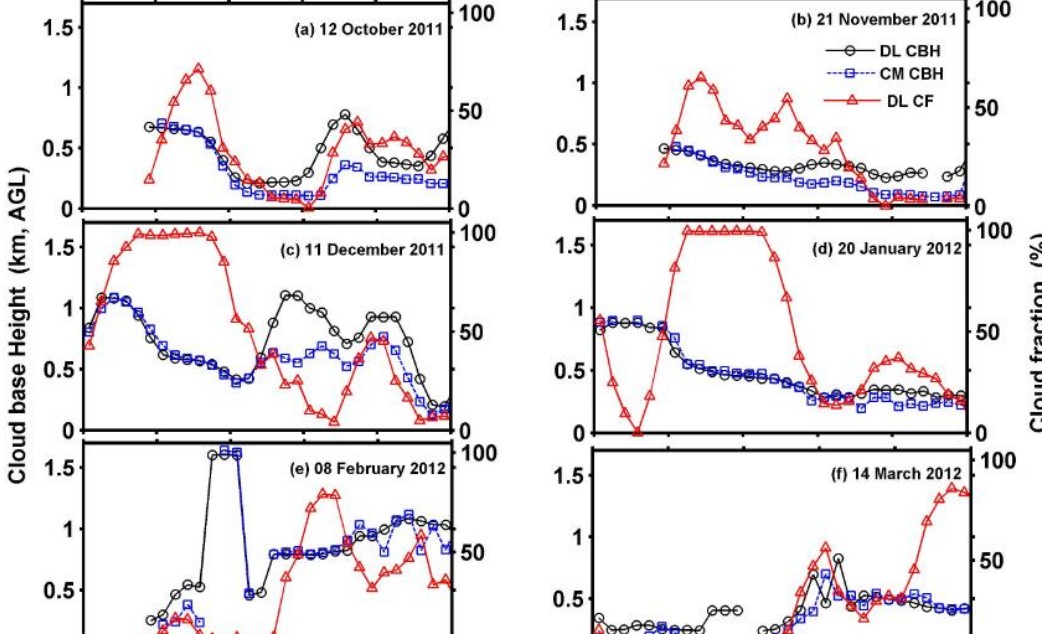

7 **Figure 5:** Temporal variation of cloud base height along with the cloud fraction observed by the Doppler Lidar

8      observed during (a) 12 October 2011, (b) 21 November 2011, (c) 11 December 2011, (d) 20 January 2012, (e) 08

9      February 2012, and (f) 14 March 2012.



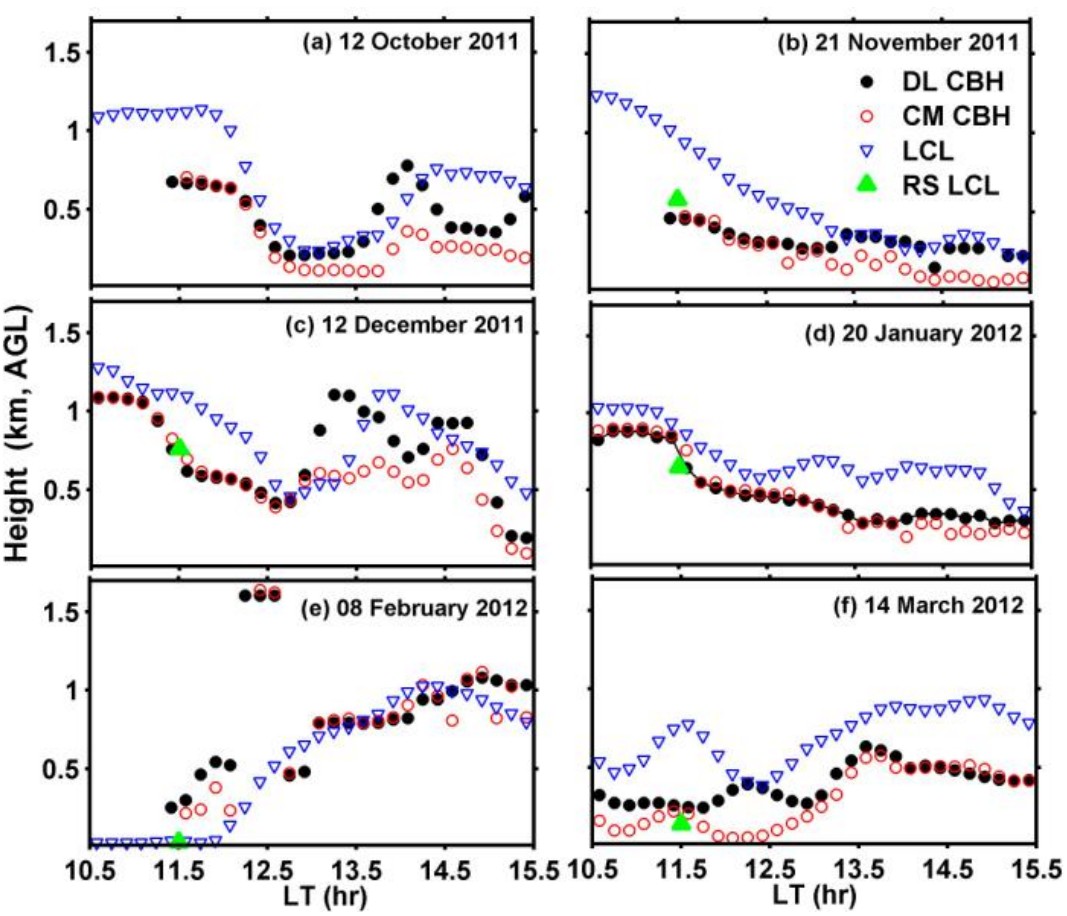

**Figure 6:** (a-f) Comparison of the cloud base height observed by Doppler Lidar and Ceilometer with lifting
condensation level (LCL) estimated by the surface meteorological parameters and Radiosonde during (a) 12
October 2011, (b) 21 November 2011, (c) 11 December 2011, (d) 20 January 2012, (e) 08 February 2012, and (f)
14 March 2012.



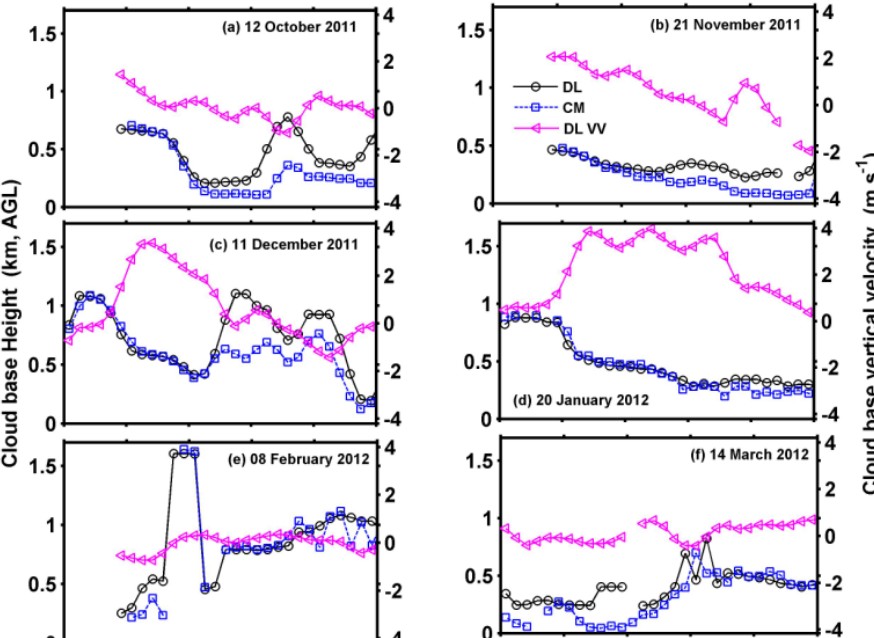

**Figure 7:** Temporal variation of CBH estimated by Doppler Lidar and Ceilometer with cloud base vertical velocity

observed by the Doppler Lidar observed during (a) 12 October 2011, (b) 21 November 2011, (c) 11 December

2011, (d) 20 January 2012, (e) 08 February 2012, and (f) 14 March 2012.



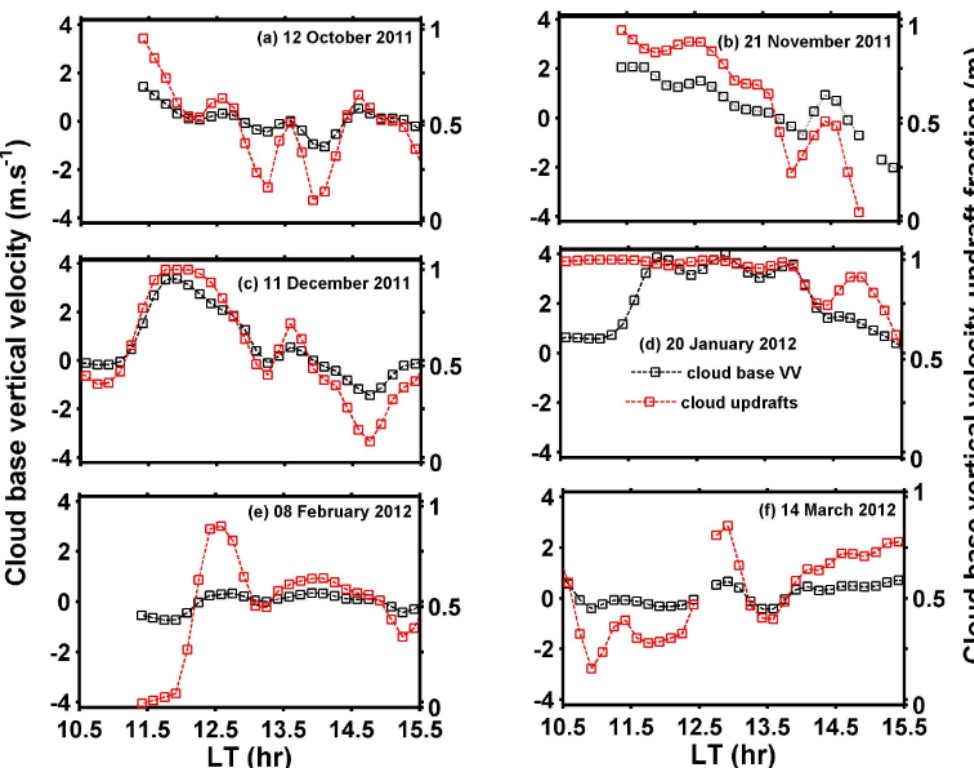

9    **Figure 8**: Temporal variation of cloud base vertical velocity along with cloud base vertical velocity updraft fraction

10    observed during (a) 12 October 2011, (b) 21 November 2011, (c) 11 December 2011, (d) 20 January 2012, (e) 08

11    February 2012, and (f) 14 March 2012.



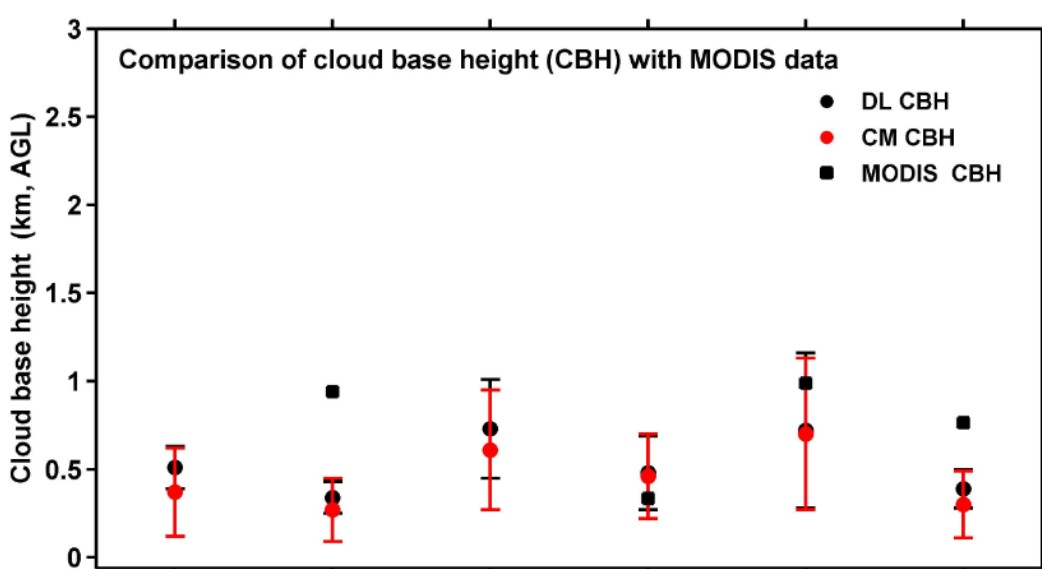

9   **Figure 9:** Comparison of cloud base height estimated by Doppler Lidar and Ceilometer during 05-10 UT and

10     MODIS Terra centered at 10.30 LT during (a) 12 October 2011, (b) 21 November 2011, (c) 11 December 2011,

11     (d) 20 January 2012, (e) 08 February 2012, and (f) 14 March 2012.







**Figure 10:** Co-relation between (a) the observed cloud base height from the Doppler Lidar and Ceilometer for all
the above six cases during 10.5-15.5 LT (hr), and (b) Cloud base vertical velocity and Cloud updraft fraction
measured by Doppler Lidar.
