# Peer review of "Identification of the cloud base height over the central 1 Himalayan region: Intercomparison of Ceilometer and Doppler 2 Lidar 3"

_Atmospheric Measurement Techniques, 2016_

## Referee Comment (RC1) · Anonymous Referee #1 · 1 Sep 2016

**General Comments**

In this manuscript Doppler Lidar measurements that were taken at a high altitude site (Manor Peak, India) during an experimental campaign that took place from June 2011 until March 2012 are used in order to calculate the cloud base height (CBH). Six periods (each one of 5 hours) were selected and were further processed based on the cloud coverage. The CBH that was calculated with a Doppler Lidar during these cases was compared against the CBH obtained from a ceilometer, and CBH from MODIS overpasses, and the Lifted Condensation Level was calculated from both radiosonde data and standard surface meteorological measurements. The vertical velocity at the CBH was also reported. The main focus of the manuscript - a method for the CBH

detection from Doppler Lidar measurements – is a novel idea, which in view of the capability of the instrument to accurately measure the wind speed, would be a valuable contribution for future studies, and certainly within the scope of the journal. One major issue is that the methods are not outlined clearly: several important details are either missing or are poorly described; and therefore the results and conclusions are not solid. Furthermore, the results and conclusions would be far more robust if more cases were studied.

Specific Comments

1. The CBH detection method from the lidar measurements should be fully described since this is the first time itis reported in the open peer-reviewed literature. Examples showing the application of the method in real SNR profiles would be helpful to that regard. In it's present form the method is briefly described and one reference (a report) is given. 2. The results would be more robust and conclusions far more convincing if more cases were included. At the moment it's not clear why only six cases are selected out of a large dataset (June 2011 – March 2012). In fact, it is even obscure how these cases were selected. It is briefly mentioned that the selection was based on the cloud coverage, residence time of clouds and availability of simultaneous datasets (P3, LN 5); rather it would be expected that the exact method by which the selection was made should be fully detailed and analytically presented in the Methodology section.

Technical Comments

P1 LN25-26: "...diurnal pattern..." It is not correct to refer to "diurnal" patterns since you only show 5 hours of measurements. P2 LN35: "...from the fair-weather ABL clouds." It is not clear at all why these are fair weather cases. P3 LN 10-11: "...and shows an increase in the pollutants level in the current climate...". Not clear what you mean. P4 LN 8-9: It would be helpful if you discuss briefly the data process used for TSI images in order i) to obtain the presence of clouds ii) to estimate the percentage of thin and opaque clouds. Probably the part of the text in Results and

[Figure]

Discussion section should be moved here. P4 LN 14: "...over a high altitude site...". Is this Manora Peak? Please state so. P4 LN 14: "DL.." Please provide details of the instrument you used (company, model, technical characteristics, etc.). P4 LN 24: "-20 dB". How was this threshold decided? Any reference? P5 LN28-29: For the current data set, the averaging interval was 30 min oversampled for every 10 Min". It is not clear what you mean that. Can you please clarify? P6 LN3-4: "Additional checks are applied to minimize false detections by rejecting temporally isolated peaks". Which tests? Please be more specific. P6, Section 3.2. How was CBH from ceilometer calculated? 1) using equation (1) or 2) from backscatter coefficient profile. P6, Section 3.3. Please be clear from the beginning that you are using the method described in Hutchinson (2002). P7 LN 26-27: "...raw...and...masked...cloud images by TSI at hourly interval...". 1) What is the difference between "masked" and "raw" images? 2) How were these images obtained from the original 30 sec images? P8 LN 14-15: "If cloud is present in the atmosphere then red pixel value is greater than where no clouds." Please rephrase. P8 LN 14-15: Stull and Eloranta, 1985 and Zhang and Klein, 2013 references are missing from the reference list. P8 LN33: "...showing high percentage of opaque cloud during 12.5-14.5 LT". This text refers to Figure 3? If so clarify. P8 LN 32-6: It's not clear how the thin/opaque cloud percentage links to the SNR or backscatter time series in Figure 4. Please be more precise. P9 LN 1-2: "It is interesting to note that the temporal evolution and duration of thin and opaque clouds in both the instruments are in reasonable agreement during all events." How thin and opaque clouds can be inferred from ceilometers and Doppler lidars? P9 LN 11-13: "Figure 6 depicts the temporal variation of CBH observed by the DL and CM along with lifted condensation level (LCL) height estimated by using surface MET parameter and RS on (a) 12 October 2011 (b) 12 21 November 13 2011 (c) 11 December 2011 (d) 20 January 2012 (e) 08 February 2012 and (f) 14 March 2012". This is a repetition of the legend of Figure 6 and could be omitted or shortened. P9 LN15: "...during the convective period...". How is the convection period defined? P9 LN 16-20: "ABL cloud heights are estimated by using LCL (Stackpole, 1967). The well-mixed ABL air parcels

which have a dry-adiabatic temperature profile and a constant mixing ratio are used to determine the LCL profile (Craven et al., 2002). For the detection of CBH, the LCL is a good approximation as the CBH depends on the relative humidity and temperature near the surface. The LCL depends on the temperature and dew point temperature above the surface and thus a good proxy for CBH.": This text could be moved to Section 3.4. P9 LN 22: "...LCL heights with the MET and RS shows a similar pattern..." In all cases RS LCL and LCL do not agree. Why? P9 LN 23-24: "From Figure 6 (a-f), it is clearly observed that in all the cases, CBH is coupled with the LCL estimated from the surface meteorological parameters.". The differences between CBH from lidar and ceilometer are not small in some cases (e.g. 12/10/2011 13.5 − 14.5) and even the trends are not always the same (e.g. 14/3/2011 11.5 − 13). P10 LN3: "...where the differences are slightly higher and need to be investigated for possible inconsistencies." Which would the reasons for these inconsistencies? P10 LN7: "R2=0.76" It is 0.81 in Figure 10. A general comment regarding the comparisonbetween CBH from lidar and ceilometer: would you expect any differences on the basis of the different operation wavelengths of the two instruments? P10 LN15: "...micropluse.." is micropulse P10 LN24: "...large scale updrafts...". More evidence would be required to support the statement that the detected updrafts are due to large scale movements. P11 LN4: "R2=0.76" same comment as above. P11 LN 13-27: Part of the text is more relevant with the "Results and Discussion" section rather than the "Summary and Conclusions". P14 LN16: Rodts et al., 2014 does not appear in the text. P14 LN21: Sarangi et al., 2014 does not appear in the text. P14 LN35: Stull, 1985 does not appear in the text. Figure 9: MODIS measurements for 12/10/2011 and 11/12/2011 can not be seen. Figure 10: Are the data points 10 minute mean means? Please provide all relevant information, Why are some values excluded from the dataset?

---

## Short Comment (SC1) · 14 Sep 2016

The paper deals mainly with the potential of a Doppler lidar (DL) to determine the cloud base height (CBH). The retrieval from a ceilometer (CM) is used as reference. There is already one review (anonymous reviewer #1) available so I only want to add a few very brief comments that were not already covered by him/her:

- A very well elaborated study comparing DL and CM has been published by Schween et al. (2014). This paper should be considered in the framework of this manuscript.

- The specific properties of the used ceilometer must be taken into account, e.g.

overlap, calibration, water vapor absorption etc.; in particular the overlap might influence the retrieval of the CBH. Note, that the CT25k provides 'backscatter' but not 'aerosol backscatter' in a quantitative way.

Moreover, there are several methodologies to retrieve the CBH developed by the manufacturer and/or the scientific community. Depending on the methodology the results might be different. Consequently, the applied algorithm should be briefly outlined or characterized by a (easy to access) publication.

- Are problems of multi-layer clouds encountered when retrieving the cloud coverage of opaque and thin clouds from the TSI? How often are several cloud layers observed by the CM? Is this an issue?

- The good agreement between the time-height cross sections from DL and CM as shown in Fig. 4 is not surprising ('It is interesting to note that the temporal...'), the authors should rather focus on a discussion of the differences. A comment how they distinguish opaque and thin clouds from Fig. 4 would be welcome (from the DL/CM or from the TSI?). What is 'Backscatter' meaning (right color code): 0.004 $km^{-1} sr^{-1}$ seems to be quite low for a thick cloud?

- When comparing the CBHs from MODIS and DL/CM (Fig. 9) differences are likely due to the different sampling (i.e., point measurement vs. spatial average over 1 × 1 degree, different temporal averaging). At least an estimate of the accuracy of MODIS-algorithm for the CBH should be added (Sec. 3.3). UT should be changed to LT (as used in the rest of the paper).

- When comparing CBHs at different sites the authors switch from height 'above ground level' to height 'above mean sea level' and found agreement with their own site (Tab. 1). Is the 'amsl' more reasonable from a meteorological point of view? Is there a 'need' that the CBH at Nainital is similar to that at, say, Lindenberg? Nobody will in principle doubt the CBHs retrieved from the CM in this paper, thus no justifcation is required.

- page 10, line 23: 'The observations from all the instruments...'. There is something wrong with this sentence.

- There are several redundancies, e.g. lines 12/13 on page 11 already appear in the previous paragraph.

Suggested references:

Schween, J. H., Hirsikko, A., Löhnert, U., and Crewell, S.: Mixing-layer height retrieval with ceilometer and Doppler lidar: from case studies to long-term assessment, Atmos. Meas. Tech., 7, 3685-3704, doi:10.5194/amt-7-3685-2014, 2014.

Wiegner, M., Madonna, F., Binietoglou, I., Forkel, R., Gasteiger, J., Geiß, A., Pappalardo, G., Schäfer, K., and Thomas, W.: What is the benefit of ceilometers for aerosol remote sensing? An answer from EARLINET, Atmos. Meas. Tech., 7, 1979-1997, doi:10.5194/amt-7-1979-2014, 2014.

Haeffelin, M., F. Angelini, Y. Morille, G. Martucci, S. Frey, G. P. Gobbi, S. Lolli, C. D. O'Dowd, L. Sauvage, I. Xueref-Remy, B. Wastine, and D. G. Feist: Evaluation of Mixing-Height Retrievals from Automatic Profiling Lidars and Ceilometers in View of Future Integrated Networks in Europe, Boundary-Layer Meteorol (2012) 143:49–75, DOI 10.1007/s10546-011-9643-z.

Münkel, C., Schäfer, K., and Emeis, S.: Adding confidence levels and error bars to mixing layer heights detected by ceilometer, Proc. SPIE 8177, 817708–1 – 817708–9, 2011 [this is a paper on a more recent Vaisala-ceilometer].

---

## Referee Comment (RC2) · Anonymous Referee #2 · 3 Oct 2016

**General comments**

Reliable detection of cloud base height is a key parameter in several scientifically and societally important applications. Ceilometers were developed for this challenging task. Performance of ceilometers, and subsequently developed retrieval algorithms, to detect cloud base heights (CBH) has been intensively investigated by several research groups. Reliability of CBH detection depends on several factors, including, used detection algorithm (Martucci et al., 2010), type of cloud hydrometeors (Van Tricht et al., 2014), sensor wavelength and likely its sensitivity (Schween et al., 2014). It is true, however, that a published comparison of a Doppler lidar (wavelength of 1.5 microm) and ceilometer (905 nm) is missing. This manuscript presents performance compari-

son of Vaisala CT25k ceilometer and Doppler lidar CBH detection which is well within scope of the Journal.

**Specific comments**

A major disadvantage of the manuscript is related to the fact that sensor performance is compared by using two different data retrieval methods. In addition to presented results, the authors should apply only one method to data from both sensors in order to reliably compare performance of the two sensors. The second major comment is related to amount of data employed in statistical analysis (Fig. 10). In addition to six days of case studies, the authors should show statistical comparison covering the entire measurement campaign, and subsequently, add figure 11 displaying CBH comparison from Doppler lidar and ceilometer. This would significantly add value to the manuscript.

**Minor and technical comments**

2.2 Doppler lidar

Page 4, lines 14-24: Add a sentence to explain how you determined SNR threshold (-20 dB), and mention model and manufacturer of the Doppler lidar. A discussion on data quality should cite at least to work by Manninen et al. (2016). It is true, however, that in this case study corrections to signal-to-noise ratio as suggested by Manninen et al. (2016) would have limited, if any, impact to presented results.

3.2 CBH retrieval using CM

Add more information on CBH retrieval or state that standard output of Vaisala CT25k has been used.

4. Results and discussion

Page 9, lines 1-2: 'It is interesting to note that the temporal evolution and duration of thin and opaque clouds in both the instruments are in reasonable agreement during all

events'. Discuss more on this topic since it is the fundamental question behind your research, i.e. why temporal evolution would be different etc.

Page 10, line 19: Table 1 does not show a detailed comparison of cloud base heights observed worldwide. In fact, I recommend removing Table 1 as it shows 8 single-day examples. To me main point of the current manuscript is not height of observed CBH, but rather, reliability of methods and sensors. Reconsider lines 15-33 on page 10.

5. Summary and conclusions

Discuss somewhere in the manuscript why you would expect, and in fact, present differences in CBHs from Doppler lidar and ceilometer.

Figures

Page 20, Fig. 4: Explain interpretation of color scales in the figure caption for the journal readership.

**References**

Manninen A.J. et al.: A generalised background correction algorithm for a Halo Doppler lidar and its application to data from Finland, Atmos. Meas. Tech., 9, 817-827, 2016.

Martucci, G. et al.: Detection of Cloud-Base Height Using Jenoptik CHM15K and Vaisala CL31 Ceilometer, J. Atmos. Ocean. Tehcnol., 27, 305-318, 2010.

Schween, J. et al.: Mixing-layer height retrieval with ceilometer and Doppler lidar: from case studies to long-term assessment, Atmos. Meas. Tech., 7, 3685-3704, 2014.

Van Tricht, K. et al.: An improved algorithm for polar cloud-base detection by ceilometer over the ice sheets, Atmos. Meas. Tech., 7, 1153-1167, 2014.

---

## Referee Comment (RC3) · Anonymous Referee #3 · 6 Oct 2016

**1  summary:**

The authors present a method to derive cloudbase height (CBH) from the profiles of signal to noise ratio (SNR) of a doppler windlidar (DL). They compare results with CBH from a ceilomter (CM) and with satellite data. Although such a comparison could be valuable the paper misses a lot of possibilities to set results in relation. It does not answer questions like "what are key differences of the retrievals", or "when and why do results deviate from each other". The description of the DL-CBH retrieval is at key points only vague and difficult to understand. The results are interpreted in a very optimistic way as 'good agreement' although large differences can be observed.

Argumentation is in many cases not straight forward but runs in circles around and is full of commonplaces where one would expect details about the observations.

Especially the statement that cumulus clouds are connected to surface processes is repeated several times without investigating when this is the case in the underlying data set. As the Doppler Lidar provides vertical velocity this could be done with e.g. the methods described in Schween et al. (2014), OConnor et al (2010) or Harvey et al. (2013).

**2  Comments in detail:**

**2.1  Title:**

The title of the paper is misleading: it states that it deals with data from a site in the central Himalaya. Data stems from the ARM mobile facility deployment at Manora Peak (1958m above mean sea level) during the Ganges Valley Aerosol Experiment (GVAX). The final campaign report (Kotamarthi 2013) locates the site in the "foothills of the Himalayan mountain range". A look into a map shows that the site is indeed at the very south-western edge of the Himalayan mountain range but not in its center.

Nevertheless the location is interesting as it lies in the sub-tropics under the influence of the Indian monsoon. The site itself provides also somewhat a challenge because it is situated on top of a mountain and local orography might influence cloud formation. But this is not even mentioned in the paper.

**2.2  1. Introduction:**

One would expect that a paper describing a new method would discuss exisiting comparable methods. In this case this would be e.g. CBH retrieval methods based on gradients in the backscatter profile as e.g. described in Martucci et al. 2010, threshold based methods as in Van Tricht et al 2014, multisensor, approaches as in Cloudnet (Illingworth et al. 2007), or visibility based concepts as is done by the Vaisala Ceilometers (see Vaisala Oyj (2002) or Morris (2012)).

**2.3  2. Observational site, instrumentation and methodology**

The climate information is given as maximum and minimum temperatures (no average, no precipiation) for the months March to May and December to February (why in this backward order ?). Unfortunately this excludes the months October and November during which two of the 6 presented cases occurred. Interesting would be also a discussion of the CBH statistics from Singh et al. (2016) (this paper shares one of the Co-authors).

**2.4  2.1. total sky imager, TSI**

page 4 line 3:

*"...we have processed the raw cloud images of the TSI."*

Does this mean that the authors made their own retrieval or do they use the retrieval of the manufacturer ? In every case there is missing a description how the differentiation between opaque and thin clouds works (as far as i remember one can adjust parameters in the manufacturers retrieval. Is this done here ?), and how cloud cover is determined (just counting pixels or by weighting parts of the image differently ?). I

see that some description appears below in section 4 "Results and discussion" which would rather belong here. And this description is also not complete.

**2.5  2.2. Doppler Lidar**

There is missing Manufacturer and type of the instrument. Does it direct Doppler measurement or is it a multi-pulse lidar ?

**2.6  2.6. MODIS**

Obviously a data product from Modis is used: which version, any reference how it works and how accurate it is?

**2.7  3.1. Cloud statistics from the DL**

This is the new algorithm presented and evaluated in this paper.

If i understand it correctly it is the same as described in an ARM report by Newsom et al. (2015) But the description here and in Newsom differ and both miss some important details:

page 5 line 32:

*"... by detecting the heights of sharp spikes in the range corrected SNR."*

Accordingly the method does not use the profile of the backscatter coefficient but instead the signal to noise ratio (SNR). How is this SNR defined? Is it the ratio between peak power in the Doppler spectrum and noise at larger Doppler shifts ? Or is it the ratio between average backscattered signal and standard deviation in the return signal of the multiple pulses ? Why must the SNR be range corrected ? - i would exepct that

the range dependence cancels in a SNR. Why is not the backscatter coefficient used ?

page line 33:

*"...the DL uses a narrow Gaussian filter..."*

What means narrow in meters or bins ? How is this value motivated ? Convolution of the signal results in a smoothed profile.which is of advantage for the following calculation of the first derivative. Newsom et al (2015) use simply the maximum of the smoothed profile. Here a pair of adjacent strong positive and negative peaks is searched, which must enclose a zero intercept. I guess one may find several in a single profile. How is the best candidate for CBH idenitified ? Is there a threshold for a 'strong peak' ? Why is the Newsom retrieval altered ?

page 6, line 2:

*"Additional checks are applied to minimize false detections by rejecting temporally isolated peaks."*

How are these Additional checks done (thresholds, range of comparison etc.) ?

**2.8 3.2. CBH retrieval by using CM**

The Basic Ranging equation (from LIDAR=LIght Detecting And Ranging) is also valid for the Doppler-Lidar described before and one may assume that the interested reader knows the LIDAR principle. More interesting would be here a description of the principle of the CBH determination in the Vaisala instrument. According to the Vaisala manual it is based on a visibility threshold. The Vaisala-CBH typically lies above the maximum in the backscatter coefficient profile as e.g. can be seen on this page of the CEILINEX intercomparison: http://ceilinex2015.de/special-topics/test. Insight in the retrieval would probably help to understand differences in the resulting CBH.

**2.9 3.3 CBH Retrieval by MODIS**

The authors use a constant liquid water content (Hess et al. 1998) which is a global average for clean air cumulus clouds. The liquid water content in Cumulus clouds typically increases with height (water condensates from rising air) and should depend on many parameters like strength of the updraft or temperature. The constant value used here is a global average with a corresponding large uncertainty. One could compare it with values derived with the remote sensing instruments suite of the AMF. Due to this and the rather large grid of the Modis retrieval ($1x1deg \simeq 1E4km^2$) one could expect that the resulting CBH has a large uncertainty. A discussion of this is missing.

**2.10 3.4 Lifted condensation level estimation by using surface MET and RS datasets**

The method described here is somehow in an accuracy-inbalance: Determination of the dew point can be written as

Td = invEsat( RH/100*Esat(T) )

with Esat(T) the water vapor saturation pressure as a function of Temperature and invEsat(e) the inverse of Esat, i.e. Dewpoint-Temperature as a function of water vapor pressure. For Esat(T) is used here the Goff-Gratch (1946) equation which is recommended by the WMO as the most accurate one (especially for very low temperatures downto -100degC). For invEsat is used in contrast hereto the inverse of the Clausius(1850)-Clapeyron(1834)-equation which is rather inaccurate as it does not consider the temperature dependence of the vaporization energy L. As a result TD at 100percent relative humidity will not be equal to air temperature.

page 7, eq 8 (Lifting condensation Level)

The Espy equation used here assumes a constant vertical gradient of the dew point dTd/dz. There are other more accurate formulas as e.g. discussed in Lawrence (2005)

or used in Stackpole (1967). Later in the text (page 9 line 16) it is stated that the method of Stackpole (1967) is used leaving the reader in uncertainty what the method in use was.

**2.11 4. Results and discussion**

How where the 6 cases selected ? Looks as if it is simple one case per month. Are these the only cloudy days ? What was the synoptic situation ?

In Fig 3 are shown time series of the cloud categorization on speparated axes with diferent scaling for thin (black, left axis) and opaque clouds (red, right axis). It would be more convenient to show time series of Popaque and the P(total)=P(opaque)+P(thin) to get an idea how much of the sky is obscured by clouds and to circumvent the wrong attribution of opaque clouds as thin clouds visible in fig 1.

Page 8, lines 12-20:

*"To classify the thin and opaque clouds, we have performed the red-green-blue ... are given in Slater et al., (2001)."*

belongs rather to section 2.1 TSI (methodology). After reading this text it is not clear whether Koehler (1991), Slater et al (2001) or Long et al.(2006) (cited on the same page above) has been used for the categorization.

Page 8, line 21:

*"The dominance of opaque clouds is clearly seen from the figure 3(a-f) during afternoon..."*

I do not agree: there are indeed more clouds in the afternoon in fig3 a, b, c, but not in the other three cases. Additionally fig 7 indicates that cloud vertical velocity is only in one case significantly higher around noon.

Page 8, Line 26 ...

*"The development of convective clouds in the lowest part of ABL is due to the presence of convective thermals ..."*

It would be great if this would be analyised with this data set: The Doppler lidar provides vertical velocity profiles and it could be clearly detected whether the observed clouds are due to rising thermals.

page 8 Line 30:

Stull and Eloranta 1985 and Zhang and Klein 2013 are missing in the references.

page 9, paragraph following line 3, discussion of figure 5 :

*"... we have also found that the frequency of occurrence of clouds is higher during afternoon."*

Again i do not agree: the cloud frequency is only higher during afternoon hours in cases e and f (Feb.8 and Mar.14) all other cases show either a decrease or complex patterns. Beside this it would be interesting to compare the 'cloud occurence frequency' from the Doppler Lidar which relies on the point measurement directly above the site with the cloud cover from the TSI which covers a larger area.

page 9, paragraph following line 11, discussion of figure 6 :

Fig. 6 shows the temporal evolution of the CBH from the two instruments together with LCL (fig 6). Differences are visible and can be large (e.g. fig6a after 13.5h LT: diff.= 500m or fig 6f before 12.5: diff>500m) but are not discussed. Instead it is stated that *"there is a strong corelation between the CBH observed by the DL and CM for all cases."* (line 13). Similar for the differences between CBH and LCL: they are in the order of several hundred meters but are not discussed.

page 9 paragraph following line 27, discussion of fig. 7:

*"From figure 7(a-f), it is clearly evident that the updrafts are dominant due to the diurnal evolution of convective ABL during daytime over the site."*

To me this is not clear: cloudbase-vertical-velocity (vv) is in most cases decreasing during the day and also shows negative values even in the afternoon (e.g. case a, case c). Again: an investigation in terms of the state of the Boundary layer considering the whole profile of the veritcal velocity (Schween et al 2014 or Harvey et al 2013) would be a great improvement.

page 9, paragraph following line 34, discussion of fig. 9:

Error bars in Fig 9 are standard deviations of CBH over the whole day ignoring that CBH shows a distinct diurnal course and thus can not be seen as an uncertainty for the time of the satellite overpass. It is stated that the agreement with the MODIS derived CBH is good. But as far as i see there are four MODIS CBH values of which only two are good in terms of the overestimated error bars. In my opinion this is not a good agreement.

page 10, paragraph starting with line 4, discussion of fig 10:

*"It is noticed that the CBH estimated by the DL is well correlated ($R^2$ =0.76)"*

This is a very positve view on the figure. It would be interesting to see parameters like root mean square error (RMSE), bias and the parameters of the linear fit shown in the figure. The plot shows that differences can be large (up to 500m), that in many cases CBH(DL) > CBH(CM) and that there is a systematic overestimation by the DL compared to the CM when CBH is below 300m. It would be important to discuss these differences in terms of the parameters which are used for the estimate (SNR versus backscatter coefficient), the method used (height of maximum versus visibility) and the state of the boundary layer (stable, turbulent, convective).

One further comment:

At several places CBH and cloud cover are named microphysical cloud properties

which in fact are rather macrophysical (see e.g. http://glossary.ametsoc.org/wiki/Cloud_microphysics).

**3 References:**

Harvey, et al. 2013, "A method to diagnose boundary-layer type using Doppler lidar, Quat. J. Roy. Meteorol. Soc., 139/676, 1681-1693, doi:10.1002/qj.2068, 2013.

Illingworth, A.J. et al. 2007, "Cloudnet - continuous evaluation of cloud profiles in seven operational models using ground-based observations"

Kotamarthi, VR 2013, "Ganges Valley Aerosol Experiment (GVAX) Final Campaign Report", DOE/SC-ARM-14-011, https://www.arm.gov/sites/amf/pgh

Lawrence 2005, "The Relationship between Relative humidity and the Dewpoint Temperature in Moist Air", BAMS pp225

Martucci, G., Milroy, C., and O'Dowd, C.: Detection of cloud-base height using Jenoptik CHM15K and Vaisala Cl31 Ceilometers, J. Atmos. Ocean. Tech., 27, 305-318, doi:10.1175/2009JTECHA1326.1, 2010

Morris V.R.: Vaisala Ceilometer (VCEIL) Handbook, 1 DOE/SC-ARM-TR-020, 2012.

Schween et al. 2014: Mixing layer height retrieval with ceilometer and Doppler lidar: from case studies to long-term assessment, Atmos. Meas. Tech., 7, 3685-3704, doi:10.5194/amt-7-3685-2014.

Van Tricht, K., et al., 2014: An improved algorithm for polar cloud-base detection by ceilometer over the ice sheets, Atmos. Meas. Tech., 7, 1153-1167, doi:10.5194/amt-7-1153-2014.

O'Connor et al., 2010, "A Method for Estimating the Turbulent Kinetic Energy Dissipation Rate from a Vertically Pointing Doppler Lidar, and Independent Evaluation from Balloon-Borne In Situ Measurements", J. Atmos. Ocean. Tech., 27

---

## Author Comment (AC1) · 17 Nov 2016

The paper deals mainly with the potential of a Doppler Lidar (DL) to determine the cloud base height (CBH). The retrieval from a ceilometer (CM) is used as reference. There is already one review (anonymous reviewer #1) available so I only want to add a few very brief comments that were not already covered by him/her:

Response: We appreciate Dr Mathias Weigner for the suggestions in the form of short comments on the manuscript.

• A very well elaborated study comparing DL and CM has been published by Schween et al. (2014). This paper should be considered in the framework of this

manuscript.

Response: "Schween et al., (2014) studied the mixing layer height (MLH) by using Doppler Lidar vertical velocity standard deviation and ceilometer aerosols backscatter as it is play an important role in the atmospheric dynamics. They have proved that ceilometer is a potential instrument for the estimation of MLH by using aerosols as proxy and also cloud base height (CBH)". We have somehow missed out and now we have included (Page-2, Lines: 17-20) the reference at the appropriate place in the revised manuscript

• The specific properties of the used ceilometer must be taken into account, e.g. overlap, calibration, water vapor absorption etc.; in particular, the overlap might influence the retrieval of the CBH. Note, that the CT25k provides 'backscatter' but not 'aerosol backscatter' in a quantitative way. Moreover, there are several methodologies to retrieve the CBH developed by the manufacturer and/or the scientific community. Depending on the methodology the results might be different. Consequently, the applied algorithm should be briefly outlined or characterized by a (easy to access) publication. Response: We have now included two references related to specific properties of CT25K ceilometers which are relevant in the modified (Page-6, Lines: 2-6) version of draft. We have also outlined the methodology used for estimating CBH with ceilometers in the manuscript.

1. Münkel, C., Eresmaa, N., Räsänen, J., and Karppinen, A.: Retrieval of mixing height and dust concentration with lidar ceilometer, Bound. Lay. Meteorol., 124, 117–128, doi: 10.1007/s10546-006-9103-3, 2007. 2. Münkel, C., Schäfer, K., and Emeis, S.: Adding confidence levels and error bars to mixing layer heights detected by ceilometer, Proc. SPIE 8177, 817708–1 – 817708–9, 2011. 3. Schween, J. H., Hirsikko, A., Löhnert, U., and Crewell, S.: Mixing-layer height retrieval with ceilometer and Doppler lidar: from case studies to long-term assessment, Atmos. Meas. Tech., 7, 3685-3704, doi: 10.5194/amt-7-3685-2014, 2014.

• Are problems of multi-layer clouds encountered when retrieving the cloud coverage of opaque and thin clouds from the TSI? How often are several cloud layers observed by the CM? Is this an issue? Response: No, we have not encountered this type of multi layer clouds issue during retrieval of thin and opaque clouds for all the examples chosen for the present study. Singh et al., (2016) observed by using Ceilometer the dominance of single layer clouds is more and only ∼10 % are multilayer clouds.

Singh, Narendra, Solanki, Raman, Ojha, N., Naja, M., Dumka, U. C., Phanikumar, D. V. Sagar, Ram, Satheesh, S. K., Moorthy, K. Krishna, Kotamarthi, V. R. and Dhaka, S. K.: Variations in the cloud-base height over the central Himalayas during GVAX: association with the monsoon rainfall. Current Science, 111, 109-116, 2016.

• The good agreement between the time-height cross sections from DL and CM as shown in Fig. 4 is not surprising ('It is interesting to note that the temporal...'), the authors should rather focus on a discussion of the differences. A comment how they distinguish opaque and thin clouds from Fig. 4 would be welcome (from the DL/CM or from the TSI?). What is 'Backscatter' meaning (right color code): 0.004 km-1 sr-1 seems to be quite low for a thick cloud?

Response: We have now mentioned those lines in the context of Doppler Lidar where there was reasonably good matching between the instruments. However, some differences are evident as the backscattering are slight different. In some cases, (November 02 and March 11) we observe some differences may be because of different wavelength of the Doppler Lidar and Ceilometer. We have modified the above mentioned discussion part as per the suggestions. The thin and opaque cloud classification is done from the TSI only. For Thick cloud, it is order of 0.01 km-1 sr-1 and it is also here (Page-11, Lines:6-9).

• When comparing the CBHs from MODIS and DL/CM (Fig. 9) differences are likely due to the different sampling (i.e., point measurement vs. spatial average over 1X1 degree, different temporal averaging). At least an estimate of the accuracy of MODIS-

algorithm for the CBH should be added (Sec. 3.3). UT should be changed to LT (as used in the rest of the paper).

Response: We have changed UT to LT in caption of figure 9 in the revised version of the draft (See Page:30, Line: 13). We have also added a reference on the accuracy of MODIS algorithm for the CBH in the revised manuscript.

• When comparing CBHs at different sites the authors switch from height 'above ground level' to height 'above mean sea level' and found agreement with their own site (Tab. 1). Is the 'amsl' more reasonable from a meteorological point of view? Is there a 'need' that the CBH at Nainital is similar to that at, say, Lindenberg? Nobody will in principle doubt the CBHs retrieved from the CM in this paper, thus no justification is required.

Response: Our main motive behind giving the values of other locations was to give a comprehensive picture about the CBH's estimation around the globe. For comparison, we have added our location altitude (from above mean sea level) and CBH altitude. No, it will never same for the Lindenberg or any location because the topography also plays an important role in the cloud formation over any site (Page-20, Line1: Table2).

• page 10, line 23: 'The observations from all the instruments...'. There is something wrong with this sentence. Response: We have modified the above sentence in revised version of manuscript (Page-12, Lines:14-15).

• There are several redundancies, e.g. lines 12/13 on page 11 already appear in the previous paragraph.

Response: We have now removed redundancies from the text in the revised manuscript..

Suggested references:

Schween, J. H., Hirsikko, A., Löhnert, U., and Crewell, S.: Mixing-layer height retrieval with ceilometer and Doppler lidar: from case studies to long-term assessment, Atmos.

Meas. Tech., 7, 3685-3704, doi: 10.5194/amt-7-3685-2014, 2014.

Wiegner, M., Madonna, F., Binietoglou, I., Forkel, R., Gasteiger, J., Geiß, A., Pappalardo, G., Schäfer, K., and Thomas, W.: What is the benefit of ceilometers for aerosol remote sensing? An answer from EARLINET, Atmos. Meas. Tech., 7, 1979-1997, doi: 10.5194/amt-7-1979-2014, 2014.

Haeffelin, M., F. Angelini, Y. Morille, G. Martucci, S. Frey, G. P. Gobbi, S. Lolli, C. D. O'Dowd, L. Sauvage, I. Xueref-Remy, B. Wastine, and D. G. Feist: Evaluation of Mixing-Height Retrievals from Automatic Profiling Lidars and Ceilometers in View of Future Integrated Networks in Europe, Boundary-Layer Meteorol (2012) 143:49–75, DOI 10.1007/s10546-011-9643-z.

Münkel, C., Schäfer, K., and Emeis, S.: Adding confidence levels and error bars to mixing layer heights detected by ceilometer, Proc. SPIE 8177, 817708–1 – 817708–9, 2011 [this is a paper on a more recent Vaisala-ceilometer.
* * *

---

## Author Comment (AC2) · 17 Nov 2016

General comments

Reliable detection of cloud base height is a key parameter in several scientifically and societally important applications. Ceilometers were developed for this challenging task. Performance of ceilometers, and subsequently developed retrieval algorithms, to detect cloud base heights (CBH) has been intensively investigated by several research groups. Reliability of CBH detection depends on several factors, including, used detection algorithm (Martucci et al., 2010), type of cloud hydrometeors (Van Tricht et al., 2014), sensor wavelength and likely its sensitivity (Schween et al., 2014). It is true, however, that a published comparison of a Doppler lidar (wavelength of 1.5 micron)

[Figure]

and ceilometer (905 nm) is missing. This manuscript presents performance compari-son of Vaisala CT25k ceilometer and Doppler lidar CBH detection which is well within scope of the Journal.

Response: We greatly appreciate the detailed review by referee #2. We have tried to address all the issues that they have raised to the maximum extent possible. These changes are implemented/modified here and also in the text. We hope these changes adequately address all the concerns raised.

Specific comments

A major disadvantage of the manuscript is related to the fact that sensor performance is compared by using two different data retrieval methods. In addition to presented results, the authors should apply only one method to data from both sensors in or-der to reliably compare performance of the two sensors. The second major comment is related to amount of data employed in statistical analysis (Fig. 10). In addition to six days of case studies, the authors should show statistical comparison covering the entire measurement campaign, and subsequently, add figure 11 displaying CBH com-parison from Doppler lidar and ceilometer. This would significantly add value to the manuscript.

Response: It is to be noted that the visibility of the atmosphere cannot be measured by the Doppler Lidar (DL) and therefore it may not yield useful results if we use same methodology for both the instruments. The method used in the present manuscript is much robust and reliable for the Doppler Lidar and hence less affected by the false detection of the CBH.

"Although, the ARM site deployment was during June 2011 – March 2012, we do not have the Doppler Lidar data during the monsoon (June – September 2011) period be-cause of washout of the aerosol particles. Moreover, we have both no/ less percentage of cloud coverage and less cloud residence time (∼1-2 hours) during other seasons. Hence, we have particularly selected those cases where we have maximum cloud cov-

erage and residence times during the daytime boundary layer (05-10 UT) convection period. Also, another aspect for selecting these cases is the availability of simultaneous datasets with other instruments like Ceilometer (CM), Radiosonde and other meteorological instruments. Some of the cases are rejected because of sudden spikes and other consistency checks. These aspects are clearly mentioned in the revised version of the manuscript" (Page4, Lines:19-27).

Minor and technical comments

2.2 Doppler lidar Page 4, lines 14-24: Add a sentence to explain how you determined SNR threshold (-20 dB), and mention model and manufacturer of the Doppler lidar. A discussion on data quality should cite at least to work by Manninen et al. (2016). It is true, however, that in this case study corrections to signal-to-noise ratio as suggested by Manninen et al. (2016) would have limited, if any, impact to presented results.

Response: We have followed the methodology described in Lenschow et al. (2000) and Pearson et al., (2009) for the SNR threshold to remove outliers. We have also provided the technical details of Doppler Lidar in Table-1 and included Manninen et al (2016) reference in Doppler Lidar section in the revised manuscript (Page-5, Lines:25-30).

Pearson, G.N., Davies, F., Collier, C.: An analysis of the performance of the UFAM pulsed Doppler Lidar for observing the boundary layer. J. Atmos. Oceanic Technol., 26, 240-250, 2009.

Lenschow D.H., Wulfmeyer, V., Senff, C.:2000 Measuring Second- through Fourth-Order Moments in Noisy Data. J. Atmos. Oceanic Tech., 17, 1330-1347, 2000.

3.2 CBH retrieval using CM Add more information on CBH retrieval or state that standard output of Vaisala CT25k has been used.

Response: The CBH estimation by CT25k is done based on visibility threshold and yes, we have used the standard output of Vaisala CT25k and these things are now

added in the revised manuscript (Page-6, Lines:2-6).

4. Results and discussion Page 9, lines 1-2: 'It is interesting to note that the temporal evolution and duration of thin and opaque clouds in both the instruments are in reasonable agreement during all events'. Discuss more on this topic since it is the fundamental question behind your research, i.e. why temporal evolution would be different etc.

Response: We meant to say that TSI thin and opaque cloud coverage is exactly matching with DL and CM cloud patterns and now typo mistake has been rectified in the revised manuscript (Page-10, Lines:22-23).

Page 10, line 19: Table 1 does not show a detailed comparison of cloud base heights observed worldwide. In fact, I recommend removing Table 1 as it shows 8 single-day examples. To me main point of the current manuscript is not height of observed CBH, but rather, reliability of methods and sensors. Reconsider lines 15-33 on page 10.

Response: Our main aim behind giving Table-1 is to summarize all the observations related to cloud base heights at one place. For comparison, we have also included observations with the different locations latitude and longitude along with CBH. This table is just for comparison with our site. As pointed out by the reviewer, our main aim is to show the potential of Doppler Lidar in the estimation of cloud base height and reciprocate with other instruments as well and now we have focused mainly on these aspects in the revised manuscript (Page-11, Lines:6-9; Lines:24-27).

5. Summary and conclusions Discuss somewhere in the manuscript why you would expect, and in fact, present differences in CBHs from Doppler lidar and ceilometer.

Response: In general, they have reasonably good agreement in most of cases, however, differences in some cases between the CBH estimated by the DL & CM may be due to working principle, retrieval techniques and also the technical specification of different instruments with different methodologies. The retrieval of signal to noise ratio with the Doppler Lidar (Hardesty et al., 1997) and backscatter from the Ceilometer (Heese et al., 2010) is also derived with different methodologies and these things are mentioned in the revised manuscript (Page-11, Lines:6-9; Lines:24-27).

Figures Page 20, Fig. 4: Explain interpretation of color scales in the figure caption for the journal readership.

Response: For better clarity, we have now explained clearly the caption of Figure 4 in the revised manuscript (Page-25, Lines:12-13).

References

Manninen A.J. et al.: A generalised background correction algorithm for a Halo Doppler lidar and its application to data from Finland, Atmos. Meas. Tech., 9, 817-827, 2016.

Martucci, G. et al.: Detection of Cloud-Base Height Using Jenoptik CHM15K and Vaisala CL31 Ceilometer, J. Atmos. Ocean. Tehcnol. 27, 305-318, 2010.

Schween, J. et al.: Mixing-layer height retrieval with ceilometer and Doppler lidar: from case studies to long-term assessment, Atmos. Meas. Tech., 7, 3685-3704, 2014.

Van Tricht, K. et al.: An improved algorithm for polar cloud-base detection by ceilometers over the ice sheets, Atmos. Meas. Tech., 7, 1153-1167, 2014.

---

## Author Comment (AC3) · 17 Nov 2016

General Comments

In this manuscript Doppler Lidar measurements that were taken at a high altitude site (Manora Peak, India) during an experimental campaign that took place from June 2011 until March 2012 are used in order to calculate the cloud base height (CBH). Six periods (each one of 5 hours) were selected and were further processed based on the cloud coverage. The CBH that was calculated with a Doppler Lidar during these cases was compared against the CBH obtained from a ceilometer, and CBH from MODIS overpasses, and the Lifted Condensation Level was calculated from both radiosonde data and standard surface meteorological measurements. The vertical velocity at the

CBH was also reported. The main focus of the manuscript - a method for the CBH detection from Doppler Lidar measurements -is a novel idea, which in view of the capability of the instrument to accurately measure the wind speed, would be a valuable contribution for future studies, and certainly within the scope of the journal. One major issue is that the methods are not outlined clearly: several important details are either missing or are poorly described; and therefore the results and conclusions are not solid. Furthermore, the results and conclusions would be far more robust if more cases were studied.

Response: We greatly appreciate the detailed review by referee #2. We have tried to address all the issues that they have raised to the maximum extent possible. These changes are implemented/modified here and also in the text. We hope these changes adequately address all the concerns raised.

Specific Comments 1. The CBH detection method from the lidar measurements should be fully described since this is the first time it is reported in the open peer-reviewed literature. Examples showing the application of the method in real SNR profiles would be helpful to that regard. In its present form the method is briefly described and one reference (a report) is given.

Response: We have now included the detailed description about the CBH estimation and also a figure showing the method for the CBH estimation by using SNR profiles for more clarity to the reader in the revised version of the manuscript.

2. The results would be more robust and conclusions far more convincing if more cases were included. At the moment it's not clear why only six cases are selected out of a large dataset (June 2011-March 2012). In fact, it is even obscure how these cases were selected. It is briefly mentioned that the selection was based on the cloud coverage, residence time of clouds and availability of simultaneous datasets (P3, LN5); rather it would be expected that the exact method by which the selection was made should be fully detailed and analytically presented in the Methodology section.

Response: Although, the ARM site deployment was during June 2011 – March 2012, we do not have the Doppler lidar data during the monsoon (June – September 2011) period because of washout of the aerosol particles. Moreover, either no/less percentage of cloud coverage or less cloud residence time (∼1-2 hours) are observed during other seasons. Hence, we have particularly selected those cases, where we have maximum cloud coverage and residence times during the daytime boundary layer (05-10 UT) convection period. Also, another aspect for selecting these cases is the availability of simultaneous datasets with other instruments like Ceilometer, radiosonde and other meteorological instruments. Some of the cases are rejected because of sudden spikes and other consistency checks. These aspects are clearly mentioned in the revised version of the manuscript

Technical Comments

P1 LN25-26: ": diurnal pattern:" It is not correct to refer to "diurnal" patterns since you only show 5 hours of measurements.

Response: Yes, agreed and we have now modified the terminology wherever appropriate in the revised manuscript.

P2 LN35: "::: from the fair-weather ABL clouds." It is not clear at all why these are fair weather cases.

Response: Yes, these cases cannot be considered as fair weather cases as they persist for much shorter timescales and now we have modified this terminology in the revised manuscript.

P3 LN 10-11: ": and shows an increase in the pollutants level in the current climate:". Not clear what you mean.

Response: Typo mistake has been corrected and replaced 'climate' by 'scenario' in the revised manuscript.

P4 LN 8-9: It would be helpful if you discuss briefly the data process used for TSI

images in order i) to obtain the presence of clouds ii) to estimate the percentage of thin and opaque clouds. Probably the part of the text in Results and Discussion section should be moved here.

Response: We have now moved some text of results and discussion to the section 2.1 in the revised manuscript for the explanation of data analysis procedure of TSI.

P4 LN 14: ": over a high altitude site:". Is this Manora Peak? Please state so.

Response: Clearly stated in the revised manuscript.

P4 LN 14: "DL." Please provide details of the instrument you used (company, model, technical characteristics, etc.).

Response: We have now added a separate table with the technical specifications of the Doppler Lidar in the revised manuscript.

P4 LN 24: "-20 dB". How was this threshold decided? Any reference?

Response: We have made a detailed analysis regarding the appropriate threshold condition for the used dataset in the current report. Based on the careful inspection of datasets from September 2011 –March 2012, and followed Lenschow et al., (2000) & Pearson et al., (2009) methodology for SNR criteria over the DL dataset and arrived at '-20 dB' is an appropriate threshold. The detailed description about the threshold methodology and corresponding references are also added in the revised manuscript.

Lenschow D.H., Wulfmeyer, V., Senff, C.:2000 Measuring Second- through Fourth-Order Moments in Noisy Data. J. Atmos. Oceanic Tech., 17, 1330-1347, 2000.

Pearson, G.N., Davies, F., Collier, C.: An analysis of the performance of the UFAM pulsed Doppler Lidar for observing the boundary layer. J. Atmos. Oceanic Technol., 26, 240-250, 2009.

P5 LN28-29: For the current data set, the averaging interval was 30 min oversampled for every 10 Min". It is not clear what you mean that. Can you please clarify?

Response: The vertical velocity and cloud statistics Value Added Product (VAP) uses a 30-minute averaging time window, but produces output using a 10-minute sampling interval. Thus, every third sample is statistically independent.

P6 LN3-4: "Additional checks are applied to minimize false detections by rejecting temporally isolated peaks". Which tests? Please be more specific.

Response: The additional checks to remove false detection are now described in more detail for better clarity in the revised manuscript. CBH estimates from profiles immediately before and after the current profile are compared and if both differences exceed 1km then the current CBH estimate is rejected.

P6, Section 3.2. How was CBH from ceilometers calculated? 1) Using equation (1) or 2) from backscatter coefficient profile.

Response: Here, we have used the visibility threshold for the detection of cloud base height by using Ceilometer and it is described in detail in Väisälä Oyj (2002) and Morris (2012)). We have also added above mentioned references in the revised version of manuscript.

P6, Section 3.3. Please be clear from the beginning that you are using the method described in Hutchinson (2002).

Response: We have now modified section 3.3 considering the suggestions in revised version of the manuscript.

P7 LN 26-27: ": raw: and:: :masked: : :cloud images by TSI at hourly interval: : :". 1) What is the difference between "masked" and "raw" images? 2) How were these images obtained from the original 30 sec images?

Response: 1) The first step toward Total Sky Imager (TSI) image analysis, TSI software masks out obstructions-the imager, its arm, and the sun-blocking band in the raw images. This is the difference between raw and masked images. 2) As instrument gives the original images at the interval of 30 sec, we have taken the original images

at every 1 hour interval.

P8 LN 14-15: "If cloud is present in the atmosphere then red pixel value is greater than where no clouds." Please rephrase.

Response: Sentence is rephrased in the revised manuscript.

P8 LN 14-15: Stull and Eloranta, 1985 and Zhang and Klein, 2013 references are missing from the reference list.

Response: References are added in the reference list in the revised manuscript.

P8 LN33: ": showing high percentage of opaque cloud during 12.5-14.5 LT". This text refers to Figure 3? If so clarify.

Response: From Figure 3 (a-b), it is clearly seen that during study period (10.5-15.5 hrs) the percentage of opaque and thin clouds shows the reverse pattern. It means that when opaque clouds are higher then thin clouds are lesser and vice-versa. Similar kinds of patterns are also observed by the Doppler Lidar and Ceilometer.

P8 LN 32-6: It's not clear how the thin/opaque cloud percentage links to the SNR or backscatter time series in Figure 4. Please be more precise.

Response: Interpretation was done by using the thin/opaque cloud percentage observed by the Total Sky Imager over the Manora Peak, when thick clouds are present overhead of Doppler Lidar and Ceilometer. It will give the high backscatter in comparison to thin clouds in both the parameters (SNR & Backscatter) which is clearly evident from the Figure 4.

P9 LN 1-2: "It is interesting to note that the temporal evolution and duration of thin and opaque clouds in both the instruments are in reasonable agreement during all events." How thin and Opaque clouds can be inferred from ceilometers and Doppler lidars?

Response: The observed temporal evolution and duration of thin and opaque clouds are observed by the Total Sky Imager (TSI) and not by the Doppler Lidar and Ceilometer. We have clarified this part more clearly in the revised manuscript.

P9 LN 11-13: "Figure 6 depicts the temporal variation of CBH observed by the DL and CM along with lifted condensation level (LCL) height estimated by using surface MET parameter and RS on (a) 12 October 2011 (b) 21 November 2011 (c) 11 December 2011 (d) 20 January 2012 (e) 08 February 2012 and (f) 14 March 2012". This is a repetition of the legend of Figure 6 and could be omitted or shortened.

Response: We have shortened the paragraph in the revised manuscript.

P9 LN15: "::: during the convective period:::". How is the convection period defined?

Response: Usually daytime we observed high vertical velocities (means positive velocity and updrafts are more dominant) which are an indicator of convection and our observational period is also during daytime and hence we have considered them as convective period.

P9 LN 16-20: "ABL cloud heights are estimated by using LCL (Stackpole, 1967). The well-mixed ABL air parcels which have a dry-adiabatic temperature profile and a constant mixing ratio are used to determine the LCL profile (Craven et al., 2002). For the detection of CBH, the LCL is a good approximation as the CBH depends on the relative humidity and temperature near the surface. The LCL depends on the temperature and dew point temperature above the surface and thus a good proxy for CBH.": This text could be moved to Section 3.4.

Response: We have now moved this paragraph to section 3.4 in the revised manuscript.

P9 LN 22: ": LCL heights with the MET and RS shows a similar pattern: : :" In all cases RS LCL and LCL do not agree. Why?

Response: In general, LCL has reasonable agreement with both the instruments, however, in some cases differences are observed due to the drift of balloon over the site and also different instruments with different methodology. We have tried to make this

explanation in a better way for better clarity.

P9 LN 23-24: "From Figure 6 (a-f), it is clearly observed that in all the cases, CBH is coupled with the LCL estimated from the surface meteorological parameters.". The differences between CBH from Doppler Lidar and ceilometer are not small in some cases (e.g. 12/10/2011 13.5 – 14.5) and even the trends are not always the same (e.g. 14/3/2011 11.5 – 13).

Response: We agree that in some cases an over estimation of $\sim$ 500 m in CBH is observed. The overestimation of the CBH by using DL in comparison to Ceilometer may due to different instruments specification and different retrieval methodologies for the CBH respectively. Moreover, for the first time we are dealing with the Doppler Lidar CBH estimation and to establish the concrete methodology, more statistical analysis would be beneficial to arrive at certain conclusions.

P10 LN3: "::: where the differences are slightly higher and need to be investigated for possible inconsistencies." Which would the reasons for these inconsistencies?

Response: Doppler Lidar also responds to higher aerosol concentration in the form of aerosol backscatter apart from cloud coverage. In some cases, higher aerosol concentration could lead to higher values in Doppler Lidar as compared to Ceilometer especially in the afternoon hours leading to forenoon-afternoon asymmetry in AOD over the Himalayan region (please refer Dumka et al., 2006; Shukla et al., 2015 for further details). Moreover, discrepancy during March 2012 may be attributed higher aerosol long range transport from middle-east locations to Indian regions which could have manifested in higher values in DL as compared to CM.

P10 LN7: "R2 =0.76" It is 0.81 in Figure 10. A general comment regarding the comparison between CBH from lidar and ceilometer: would you expect any differences on the basis of the different operation wavelengths of the two instruments?

Response: Typo mistake in the correlation values has now been rectified in Figure 11

in the revised manuscript. During observations, both the Lidars will see the clouds at the same height but the minor differences in the cloud base height might be due to their retrieval algorithms and assumptions involved.

P10 LN15: ":::: micropluse.." is micropulse

Response: Implemented

P10LN24: ": :large scale updrafts: : :". More evidence would be required to support the statement that the detected updrafts are due to large scale movements.

Response: We apologize for the typo mistake, what we mean large scale updrafts are about the higher magnitude of updrafts observed in DL during the event. We have modified the sentence in the revised manuscript.

P11 LN4: "R2=0.76" same comment as above.

Response: Corrected

P11 LN 13-27: Part of the text is more relevant with the "Results and Discussion" section rather than the "Summary and Conclusions".

Response: We have now rearranged these two sections in the revised manuscript.

P14 LN16: Rodts et al., 2014 does not appear in the text.

Response: We have removed Rodts et al reference from the reference list in the revised manuscript.

P14 LN21: Sarangi et al., 2014 does not appear in the text.

Response: We missed it somehow and now included Sarangi et al reference in section 2 in the revised manuscript.

P14 LN35: Stull, 1985 does not appear in the text. Figure 9: MODIS measurements for 12/10/2011 and 11/12/2011 can not be seen. Figure 10: Are the data points 10-minute mean means? Please provide all relevant information, why are some values excluded

from the dataset?

Response: We missed it and now we have included Stull reference in the introduction section of the revised manuscript. We do not have the MODIS over pass over our site at the above mentioned dates (October & December 2011) that is why data on those two days are not available. Yes, data points are at every 10 min interval and we did not exclude the data points in correlation plot. It is clearly seen from the figure 5 that some data gaps are observed in the temporal variation of the CBH with the Doppler Lidar and same is replicated in the correlation plots.

---

## Author Comment (AC4) · 17 Nov 2016

1. Summary:

The authors present a method to derive cloud base height (CBH) from the profiles of signal to noise ratio (SNR) of a Doppler wind lidar (DL). They compare results with CBH from a ceilomter (CM) and with satellite data. Although such a comparison could be valuable the paper misses a lot of possibilities to set results in relation. It does not answer questions like "what are key differences of the retrievals", or "when and why do results deviate from each other". The description of the DL-CBH retrieval is at key points only vague and difficult to understand. The results are interpreted in a very optimistic way as 'good agreement' although large differences can be observed.

Argumentation is in many cases not straight forward but runs in circles around and is full of commonplaces where one would expect details about the observations.

Especially the statement that cumulus clouds are connected to surface processes is repeated several times without investigating when this is the case in the underlying data set. As the Doppler Lidar provides vertical velocity this could be done with e.g. the methods described in Schween et al. (2014), O'Connor et al (2010) or Harvey et al. (2013).

Response: We greatly appreciate the detailed review by referee #3. We have tried to address all the issues that they have raised to the maximum extent possible. These changes are implemented/modified here and also in the text. We hope these changes adequately address all the concerns raised.

2 Comments in detail:

2.1 Title: The title of the paper is misleading: it states that it deals with data from a site in the central Himalaya. Data stems from the ARM mobile facility deployment at Manora Peak (1958m above mean sea level) during the Ganges Valley Aerosol Experiment (GVAX). The final campaign report (Kotamarthi 2013) locates the site in the "foothills of the Himalayan mountain range". A look into a map shows that the site is indeed at the very south-western edge of the Himalayan mountain range but not in its center. Nevertheless the location is interesting as it lies in the sub-tropics under the influence of the Indian monsoon. The site itself provides also somewhat a challenge because it is situated on top of a mountain and local orography might influence cloud formation. But this is not even mentioned in the paper.

Response: Title of paper is modified by considering the relevance of the study and also field campaign report (Kotamarthi, 2013). Importance of site in introduction section-1 (Page-3, Lines: 5-9) and in site details in section-2 are also clearly discussed in the revised manuscript.

2.2 1. Introduction: One would expect that a paper describing a new method would discuss existing comparable methods. In this case this would be e.g. CBH retrieval methods based on gradients in the backscatter profile as e.g. described in Martucci et al. 2010, threshold based methods as in Van Tricht et al 2014, multisensor, approaches as in Cloudnet (Illingworth et al. 2007), or visibility based concepts as is done by the Vaisala Ceilometers (see Vaisala Oyj (2002) or Morris (2012)).

Response: We have now considered all the suggested references described by different retrievals of the cloud base height in the introduction section (Page-2, Lines:15-20; Lines:29-36) in the revised version of manuscript.

2.3 2. Observational site, instrumentation and methodology

The climate information is given as maximum and minimum temperatures (no average, no precipitation) for the months March to May and December to February (why in this backward order?). Unfortunately, this excludes the months October and November during which two of the 6 presented cases occurred. Interesting would be also a discussion of the CBH statistics from Singh et al. (2016) (this paper shares one of the Co-authors).

Response: We have modified the sentences included with all the above details and Singh et al (2016) is cited and also added in the text in the revised manuscript (See Page-4, Lines:09-16).

Singh, Narendra, Solanki, Raman, Ojha, N., Naja, M., Dumka, U. C., Phanikumar, D. V. Sagar, Ram, Satheesh, S. K., Moorthy, K. Krishna, Kotamarthi, V. R. and Dhaka, S. K.: Variations in the cloud-base height over the central Himalayas during GVAX: association with the monsoon rainfall. Current Science, 111, 109-116, 2016.

2.4 2.1. Total sky imager, TSI

Page 4 line 3: "...we have processed the raw cloud images of the TSI."

Does this mean that the authors made their own retrieval or do they use the retrieval

of the manufacturer? In every case there is missing a description how the differentiation between opaque and thin clouds works (as far as i remember one can adjust parameters in the manufacturer's retrieval. Is this done here?), and how cloud cover is determined (just counting pixels or by weighting parts of the image differently?). I see that some description appears below in section 4 "Results and discussion" which would rather belong here. And this description is also not complete.

Response: Yes, the retrieval method of the manufacturer is utilized and fine tuning for adjusting parameters is done based on our location preferences. The detail description and references followed to determine the cloud cover is given in Morris, (2005). Description part is also modified in the revised manuscript considering all the suggestions (Page-5, Lines:07-13). "The TSI sky filter thresholds are defined as follows: Clear/Thin determines the ratio of thin cloud cover to clear sky. Thin/Opaque determines the ratio of thin cloud cover to opaque cloud cover. The values are assigned upon initial configuration of the TSI by adjusting each ratio to match the cloud in observed images". The description is moved from the "results and discussion" section 4 (Page-5, Lines: 05-07) to the section 2.1.

2.5 2.2. Doppler Lidar

There is missing Manufacturer and type of the instrument. Does it direct Doppler measurement or is it a multi-pulse Lidar?

Response: The manufacturer details are now added as a separate Table (See Page-19) with full technical specifications of the instrument.

2.6 2.6. MODIS Obviously a data product from Modis is used: which version, any reference how it works and how accurate it is?

Response: MODIS level-3 (MOD08_D3.051) data for the CBH retrieval is utilized and the references (Kishcha et al., 2007 and Platnick et al., 2015) describing the MODIS data details are added in the revised paper (See Page-6, Lines:29-30; Lines:32-33).

Kishcha, P., B. Starobinets, and P. Alpert (2007), Latitudinal variations of cloud and aerosol optical thickness trends based on MODIS satellite data, Geophys. Res. Lett., 34, L05810, doi: 10.1029/2006GL028796.

Platnick, Steven, Michael D. King, Kerry G. Meyer, Gala Wind, Nandana Amarasinghe, Benjamin Marchant, G. Thomas Arnold, Zhibo Zhang, Paul A. Hubanks, Bill Ridgway, Jéróme Riedi.: MODIS Cloud Optical Properties: User Guide for the Collection 6 Level-2 MOD06/MYD06 Product and Associated Level-3 Datasets. Version-1,http://modis-atmos.gsfc.nasa.gov/_docs/C6MOD06OPUserGuide.pdf, 2015.

2.7 3.1. Cloud statistics from the DL

This is the new algorithm presented and evaluated in this paper. If i understand it correctly it is the same as described in an ARM report by Newsom et al. (2015) but the description here and in Newsom differs and both miss some important details:

Response: Yes, Newsom et al., (2015) algorithm is followed and the estimation methodology of CBH is described in detail in the revised version (See Page-7, Lines:3-5;7-22) of manuscript and also included Figure 4 (See Page-24) for better clarity to the reader.

Page 5 line 32:

"... by detecting the heights of sharp spikes in the range corrected SNR."

Accordingly, the method does not use the profile of the backscatter coefficient but instead the signal to noise ratio (SNR). How is this SNR defined? Is it the ratio between peak power in the Doppler spectrum and noise at larger Doppler shifts? Or is it the ratio between average backscattered signal and standard deviation in the return signal of the multiple pulses? Why must the SNR be range corrected? - I would expect that the range dependence cancels in a SNR. Why is not the backscatter coefficient used?

Response: SNR is defined as mean signal power in the Doppler spectrum divided by mean noise power (Rye and Hardesty, 1997)). The attenuated backscatter is estimated

by using SNR profile by considering the telescope function and the detailed description about the methodology is given by Hirsikko et al., (2014).

Page line 33:

"...the DL uses a narrow Gaussian filter..."

What means narrow in meters or bins? How is this value motivated? Convolution of the signal results in a smoothed profile. Which is of advantage for the following calculation of the first derivative? Newsom et al (2015) use simply the maximum of the smoothed profile. Here a pair of adjacent strong positive and negative peaks is searched, which must enclose a zero intercept. I guess one may find several in a single profile. How is the best candidate for CBH identified? Is there a threshold for a 'strong peak'? Why is the Newsom retrieval altered?

Response: The Newsom retrieval method has been altered for our requirement. As stated in the revised (See Page-7, Lines:7-22) manuscript, the magnitude of the extrema in the d(r2SNR)/dr profile must exceed 0.1 km and this filters out many potential peaks and dips. We then look for the lowest peak with a corresponding dip that's between 2 and 15 range bins above the peak. If one exists, then the maximum value of r2SNR between the peak and the dip. This determines the (lowest) CBH. Alternatively, we have simply find the maximum value of the entire r2SNR profile, but we found that method results in too much false detection. The derivative technique described in this paper works better because it helps to suppress the range-dependence of the background signal.

Page 6, line 2:

"Additional checks are applied to minimize false detections by rejecting temporally isolated peaks."

How this Additional are checks done (thresholds, range of comparison etc.)?

Response: The additional checks to remove false detection are now described in more

detail in the revised (See Page-7, Lines:7-22) manuscript. CBH estimates from profiles immediately before and after the current profile are compared and rejected iIf both differences exceed 1km then the current CBH estimate.

**2.8 3.2. CBH retrieval by using CM**

The Basic Ranging equation (from LIDAR=Light Detecting and Ranging) is also valid for the Doppler-Lidar described before and one may assume that the interested reader knows the LIDAR principle. More interesting would be here a description of the principle of the CBH determination in the Vaisala instrument. According to the Vaisala manual it is based on a visibility threshold. The Vaisala-CBH typically lies above the maximum in the backscatter coefficient profile as e.g. can be seen on this page of the CEILINEX intercomparison: http://ceilinex2015.de/special-topics/test. Insight in the retrieval would probably help to understand differences in the resulting CBH.

Response: We have used the visibility threshold for the detection of cloud base height (Väisälä Oyj (2002) or Morris (2012)) and these references are included in the revised manuscript (See Page-7, Lines:35-37; Page-8, Lines:1-3).

**2.9 3.3 CBH Retrieval by MODIS**

The authors use constant liquid water content (Hess et al. 1998) which is a global average for clean air cumulus clouds. The liquid water content in Cumulus clouds typically increases with height (water condensates from rising air) and should depend on many parameters like strength of the updraft or temperature. The constant value used here is a global average with a corresponding large uncertainty. One could compare it with values derived with the remote sensing instruments suite of the AMF. Due to this and the rather large grid of the Modis retrieval (1x1deg ' 1E4km2) one could expect that the resulting CBH has a large uncertainty. A discussion of this is missing.

Response: We have now included a detailed discussion on observed differences in the derived CBH with MODIS, DL and CM. However, in the present manuscript main point

is to show the capability of DL in CBH estimation. A comparison with CM and other ground based and satellite instruments is only to ascertain that how good our CBH estimation with DL is able to represent CBH with CM and MODIS and to our surprise, values with different instruments are matching well in most of cases. However, minor discrepancies observed in some of the cases can be understood by considering the complexity of the location presented and DL CBH estimation done for the first time (Page-11, Lines:23-25).

**2.10 3.4 Lifted condensation level estimation by using surface MET and RS datasets**

The method described here is somehow in an accuracy-imbalance: Determination of the dew point can be written as Td = invEsat(RH/100*Esat(T) ) with Esat(T) the water vapor saturation pressure as a function of Temperature and invEsat(e) the inverse of Esat, i.e. Dew point-Temperature as a function of water vapor pressure. For Esat (T) is used here the Goff-Gratch (1946) equation which is recommended by the WMO as the most accurate one (especially for very low temperatures downto -100degC). For invEsat is used in contrast here to the inverse of the Clausius (1850)-Clapeyron (1834)-equation which is rather inaccurate as it does not consider the temperature dependence of the vaporization energy L. As a result, TD at 100 percent relative humidity will not be equal to air temperature.

Page 7, eq 8 (Lifting condensation Level)

The Espy equation used here assumes a constant vertical gradient of the dew point dTd/dz. There are other more accurate formulas as e.g. discussed in Lawrence (2005) or used in Stackpole (1967). Later in the text (page 9 line 16) it is stated that the method of Stackpole (1967) is used leaving the reader in uncertainty what the method in use was.

Response: We have calculated dew point temperature estimated by using equation (11) given in Lawrence (2005). However, in order to avoid confusion, we have now modified the sentences for reader's clarity in the revised manuscript (Page-9, Lines:2-

3).

2.11 4. Results and discussion

How where the 6 cases selected? Looks as if it is simple one case per month. Are these the only cloudy days? What was the synoptic situation? In Fig 3 are shown time series of the cloud categorization on separated axes with different scaling for thin (black, left axis) and opaque clouds (red, right axis). It would be more convenient to show time series of Popaque and the P(total)=P(opaque)+P(thin) to get an idea how much of the sky is obscured by clouds and to circumvent the wrong attribution of opaque clouds as thin clouds visible in fig 1.

Response: Indeed, it was very difficult to choose cases in winter because maximum no of days are clear sky. However, we could find some cloudy cases during daytime but unable to get simultaneous datasets with other instruments for other more cases. Hence, considering all the above criteria we could find only 6 cases in those 4 months. We have also presented synoptic conditions of the observational period and modified the TSI description as suggested by the reviewer in the revised manuscript (Page-4, Lines:19-27). As suggested, Ttotal cloud cover with TSI is plotted and shown in Figure-1in seperate file.

Page 8, lines 12-20:

"To classify the thin and opaque clouds, we have performed the red-green-blue ... are given in Slater et al., (2001)."

belongs rather to section 2.1 TSI (methodology). After reading this text it is not clear whether Koehler (1991), Slater et al (2001) or Long et al. (2006) (cited on the same page above) has been used for the categorization.

Response: We have moved the suggested text in the section 2.1 in the revised version of the manuscript. We have also removed the sentences with Koehler (1991) and Slater et al., 2001 in the revised manuscript which creates the confusion to the reader. The

detailed description about the classification is given in Long et al., (2006) which is used by manufacture of the TSI also (Page-5, Lines:4-12).

Page 8, line 21:

"The dominance of opaque clouds is clearly seen from the figure 3(a-f) during afternoon..." I do not agree: there are indeed more clouds in the afternoon in fig3 a, b, c, but not in the other three cases. Additionally, fig 7 indicates that cloud vertical velocity is only in one case significantly higher around noon.

Response: We meant to say here that opaque clouds are more as compared to thin clouds and we have modified the sentence in the revised manuscript (Page-10, Lines:5-6). Page 8, Line 26... "The development of convective clouds in the lowest part of ABL is due to the presence of convective thermals ..."

It would be great if this would be analyised with this data set: The Doppler lidar provides vertical velocity profiles and it could be clearly detected whether the observed clouds are due to rising thermals.

Response: We have checked the vertical velocity to confirm the rising thermals and found that the updrafts are more dominant for all cases during daytime.

Page 8 Line 30:

Stull and Eloranta 1985 and Zhang and Klein 2013 are missing in the references.

Response: We missed it somehow and now included both the references in the revised manuscript (Page-18, Lines:1-2, Lines:22-24).

Page 9, paragraph following line 3, discussion of figure 5:

"... we have also found that the frequency of occurrence of clouds is higher during afternoon."

Again i do not agree: the cloud frequency is only higher during afternoon hours in cases

e and f (Feb.8 and Mar.14) all other cases show either a decrease or complex patterns. Beside this it would be interesting to compare the 'cloud occurrence frequency' from the Doppler Lidar which relies on the point measurement directly above the site with the cloud cover from the TSI which covers a larger area.

Response: We have now plotted the monthly mean diurnal variation of cloud occurrence frequency by using Doppler Lidar which is given below in the Figure. From figure, it is clearly seen that it is showing different nature in forenoon and afternoon in all the months. During month of October and November, initially it is higher and decreasing trend up to 20.5 hrs and then showing increasing trends. The occurrence of clouds is more in the afternoon in comparison to forenoon during December. In January, it shows higher in forenoon and in the rest of day varies between 30-60%. It is showing increasing and decreasing trend in February and magnitude varies between 30-50%. As Doppler Lidar is giving first cloud base height. Therefore, we have compared the frequency of occurrence with the Singh et al., 2016 and found that an overestimation in all months (October ($\sim$3%), November ($\sim$ 2%), December ($\sim$7%), January ($\sim$ 5 %), February ($\sim$ 2 %) and March ($\sim$ 6 %)). The overestimation of cloud occurrence frequency by Doppler Lidar varies between $\sim$ 1-8 % which could be due to the different techniques and estimation method (Figure-2 in separate file).

Page 9, paragraph following line 11, discussion of figure 6:

Fig. 6 shows the temporal evolution of the CBH from the two instruments together with LCL (fig 6). Differences are visible and can be large (e.g. fig6a after 13.5h LT: diff. = 500m or fig 6f before 12.5: diff>500m) but are not discussed. Instead it is stated that "there is a strong correlation between the CBH observed by the DL and CM for all cases." (Line 13). Similar for the differences between CBH and LCL: they are in the order of several hundred meters but are not discussed.

Response: We have observed an overestimation of the CBH by using DL in comparison to CM which could be due to different retrieval techniques and technical specifications

of both the instruments. Similarly, we have also observed the difference between LCL and derived CBH in some cases but they are reasonably in good agreement in most of the cases. We have modified the revised manuscript considering all these suggestions (Page-11, Line:6-9).

Page 9 paragraph following line 27, discussion of fig. 7: "From figure 7(a-f), it is clearly evident that the updrafts are dominant due to the diurnal evolution of convective ABL during daytime over the site."

To me this is not clear: cloud base-vertical-velocity (vv) is in most cases decreasing during the day and also shows negative values even in the afternoon (e.g. case a, case c). Again: an investigation in terms of the state of the Boundary layer considering the whole profile of the vertical velocity (Schween et al., 2014 or Harvey et al., 2013) would be a great improvement.

Response: We have now modified the revised manuscript for better clarity by taking boundary layer vertical velocity into account (Page-11, Lines:13-14).

Page 9, paragraph following line 34, discussion of fig. 9:

Error bars in Fig 9 are standard deviations of CBH over the whole day ignoring that CBH shows a distinct diurnal course and thus cannot be seen as an uncertainty for the time of the satellite overpass. It is stated that the agreement with the MODIS derived CBH is good. But as far as i see there are four MODIS CBH values of which only two are good in terms of the overestimated error bars. In my opinion this is not a good agreement.

Response: We have now modified the revised manuscript by clearly stating the similarities and discrepancies observed for each case (Page-11, Line:23-25).

Page 10, paragraph starting with line 4, discussion of fig 10:

"It is noticed that the CBH estimated by the DL is well correlated ($R^2$ =0.76)"

This is a very positive view on the figure. It would be interesting to see parameters like root mean square error (RMSE), bias and the parameters of the linear fit shown in the figure. The plot shows that differences can be large (up to 500m), that in many cases CBH (DL) > CBH (CM) and that there is a systematic overestimation by the DL compared to the CM when CBH is below 300m. It would be important to discuss these differences in terms of the parameters which are used for the estimate (SNR versus backscatter coefficient), the method used (height of maximum versus visibility) and the state of the boundary layer (stable, turbulent, convective).

Response: The observed differences between the CBH DL and CM around 500m in some cases can be attributed to an overestimation of CBH by DL in comparison to CM. These differences may arise be due to different methodologies for the estimation of CBH with both the instruments. It should also be noted that minor discrepancies are observed, however, while considering the complexity of the location presented and DL CBH estimation done for the first time. ABL also plays an important role in the formation of cloud and all the cases are during convective boundary layer condition and it this may be one of the reason behind the overestimation with the DL in some cases (See Page-11, Line:6-9 and Page-13, Lines:15-21).

One further comment:

At several places CBH and cloud cover are named microphysical cloud properties which in fact are rather macrophysical (see e.g. http://glossary.ametsoc.org/wiki/Cloud_microphysics).

Response: Microphysical is modified to Macrophysical in the revised manuscript (Page-2, Line:3).

[Figure]

**Fig. 1.**

[Figure]

**Fig. 2.**

**Supplement:**

**Identification of the cloud base height over the Himalayan mountain range: Intercomparison of Ceilometer and Doppler Lidar**

4 5

6

7 8 K.K. Shukla1, 2, K. Niranjan Kumar3, D.V. Phanikumar1, Rob K Newsom4, V. R. Kotamarthi5 Taha B.M.J. Ouarda3, 6 and M. Venkat Ratnam7

9 1Aryabhatta Research Institute of observational sciences, Nainital, India

10 2Pt. Ravishankar Shukla University, Raipur, Chhattisgarh, India

- 11 3Institute Center for Water and Environment, Masdar Institute of Science and Technology, Abu Dhabi, United Arab
- 12 Emirates
- 13 4Pacific Northwest National Laboratory, Richland, Washington, USA
- 5Argonne National Laboratory, Argonne, Illinois, USA
- 14 6INRS-ETE, Quebec City (Qc), Canada
- 15 7National Atmospheric Research Laboratory, Gadanki, Tirupati, India
- 16

17 Correspondence to: D V Phanikumar (phani@aries.res.in; astrophani@gmail.com)

18

19 Abstract. We present the measurement of cloud base height (CBH) derived from the Doppler Lidar (DL), 20 Ceilometer (CM) and Moderate Resolution Imaging Spectroradiometer (MODIS) satellite over a high altitude 21 station in the Himalayan mountain range region for the first time. We analyzed six cases of cloud overpass during 22 the daytime convection period by using the cloud images captured by Total Sky Imager (TSI). The occurrence of 23 thick clouds (> 50%) over the site is more frequent than thin clouds (

19 Although, the ARM deployment over the site was carried out during June 2011-March 2012, we do 20 not have the DL data during the monsoon (June-September 2011) period because of washout of the aerosol 21 particles over the Manora Peak, Nainital. Moreover, we have both no/less percentage of cloud coverage and 22 less cloud residence time (~1-2 hours) during other seasons. Hence, we have particularly selected those cases 23 where cloud coverage and residence times are maximum during the daytime (05-10 UT) which is mostly the 24 convection dominant period. It is also to be noted the complexity in deriving CBH with the DL which is first 25 of its kind over the Himalayan region. Also, another aspect for selecting these cases is the availability of 26 simultaneous datasets with other instruments like CM, Radiosonde and other meteorological instruments. 27 Some of the cases are rejected because of sudden spikes and other consistency checks.

**28 2.1 Total Sky Imager (TSI)**

The TSI is manufactured by Yankee Environmental Systems (YES), and is commercialized version of the Hemispheric Sky Imager prototype (Long et al., 2006). The TSI-660 was deployed by AMF1 over the Manora Peak, Nainital during the GVAX to capture the cloud images during the daytime. The sky cloud images captured by TSI are 24-bit color JPEG images at 650x480 pixel resolution. TSI captures the cloud image at every 30 sec during daytime. In order to retrieve cloud information, we have used the processed raw cloud images and also retrieved cloud parameters by the manufacture of the TSI. Sky cover retrieval from TSI images is valid only for solar elevation angles >30 (zenith angles < 800) and images are processed for a 1600 field of view, ignoring the 100 of sky

1 near the horizon. It has a sun-blocking strip mask, which represents the location of the sun with a yellow dot in the 2 image. We have used TSI images to infer the presence of clouds over the site for a subsequent CBH estimate. The 3 TSI observations of cloud images are also utilized for the estimation of percentage of thin and opaque clouds over 4 the site. The scattering of blue light is more than red in clear skies and no aerosols conditions (i.e. molecular 5 scattering). The red pixel values of the TSI images are much higher in the presence of clouds over the imager 6 in comparison to no cloud conditions. Clear/Thin determines the ratio of thin cloud cover to clear sky. 7 Thin/Opaque determines the ratio of thin cloud cover to opaque cloud cover. The values are assigned upon 8 initial configuration of the TSI by adjusting each ratio to match the cloud in observed images. We prefer to 9 assign these values during mixed-phase clouds near solar noon. We have used the data in the current study 10 which is retrieved by the manufacture by using standard methods for the TSI. Detailed discussion about the 11 estimation of cloud properties by using TSI images were given in previous reports (Long et al., 2001, 2006; 12 Morris, 2005).

**13 2.2 Doppler Lidar**

14

15 DL was operated over a high altitude site Manora Peak, Nainital to measure the temporal and altitude resolved 16 vertical velocity and attenuated backscatter. Detailed descriptions of the technical characteristics of the DL are 17 given in Table-1. In order to retrieve the radial velocity by using Doppler principle, DL uses aerosols as tracer in 18 the atmosphere to observe the Doppler shift. The influence of insects or pollen is less in the DL observations 19 because the small aerosols in the background dominate the signal. The DL uses an eye-safe laser of wavelength  $\sim 1.5$ 20 um. It provides the vertical velocity and attenuated backscatter at a spatial resolution of ~30m and a temporal 21 resolution of 1 sec. The DL can scan the atmosphere in different modes (i.e. vertically Fixed-Beam Stare (FPT), 22 Range-Height Indicator (RHI) scan and Plan-Position Indicator (PPI) scan mode). The RHI and PPI scan modes are 23 known as the elevational and azimuthal scan of the atmosphere, respectively. A detailed technical description of the 24 DL system can be found in previous studies (Pearson et al., 2009; Newsom, 2012; Shukla et al., 2014). In the current 25 study, the vertically fixed-beam stare mode of the DL is used to estimate CBH. To minimize/remove random noise 26 fluctuations in the DL data, a threshold on the signal to noise ratio (SNR) of -20 dB is applied. In order to find 27 an appropriate SNR threshold for DL dataset, we have followed the methodology described in detail in 28 Lenschow et al., (2000) and Pearson et al., (2009). Additional data error analysis explanation can be found in 29 Newsom et al., 2015. The reduction in the SNR threshold values also lead to an increase of 50% in the data 30 accessibility (Manninen et al. 2016).

31

**32 2.3 Laser Ceilometer**

33

The Väisälä laser Ceilometer (CT25K) is deployed over the site for precise measurements of the CBH, vertical visibility and vertical profile of aerosol backscatter during GVAX (Väisälä Oyj, 2002). It has an eye-safe laser

36 source of wavelength ~905 nm. It provides the information at a temporal resolution of 16 sec and a spatial resolution

of 30 m in the atmosphere (Morris, 2012). The 16 sec interval data is aggregated to 1 min for better comparison with the DL. The detailed description of the technical properties of the CT25K Ceilometer which was used by various investigators in the past is available in Münkel et al., 2007, 2011; Haeffelin et al., 2012; Schween et al., 2014 and Wiegner et al., 2014. Weigner et al., (2014) showed that the observed aerosol backscatter by using CM have significant error due to various sources i.e. 10% due to change in calibration constant of the CM, ~ 20 % due to water vapor distribution in the atmosphere.

7

**8 2.4 Surface Meteorology System**

9

The in-situ sensors are used to measure the surface temperature (T), relative humidity (RH), pressure, wind speed and wind direction by the ARM surface meteorology systems (MET). The in-situ sensors are installed at specific standard heights for measurement of meteorological parameters (i.e. T & RH at 2 m; Barometric pressure at 1 m and wind speed and direction at 10 m) (Ritsche and Prell, 2011). The MET sensors provide the data at a temporal resolution of 1-min and we have averaged for 10-min from 1-min data to calculate the lifted condensation level (LCL) for comparison with the CBH of DL and CM, respectively.

16

**17 2.5 Radiosonde**

18

Väisälä Radiosondes (RS-92) were launched during GVAX at 00, 06, 12 and 18 UT daily regularly. The profiles of atmospheric parameters (temperature, relative humidity and winds) are measured by the Radiosonde (RS) at a vertical resolution of 10 m as the ascent rate of balloon is 5 ms-1 and transmitter time resolution is 2 sec. In the current study, we have used the 06 UT (11.5 hr LT) data of RS to calculate the LCL for all the cloud cases. The detailed description about the RS can be found in previous reports (Holdridge et al., 2011; Shukla et al., 2014).

24 25

**2.6 Moderate Resolution Imaging Spectroradiometer**

26

27 In addition to the ground based remote sensing techniques used for the estimation of CBH, we have also utilized the 28 MODIS satellite derived CBH over the observational site. The MODIS Terra data is obtained for the same cases as 29 measured by the ground based remote sensing instruments. We have used the MODIS level 3 (MOD08\_D3.051) 30 data in the current study. However, the spatial resolution of MODIS cloud data is of  $1^0 \times 1^0$  latitude-longitude 31 grids. We have used the cloud top pressure, cloud optical depth and effective radius of liquid cloud for all the cases. 32 A detail description of the MODIS data is given for instance in Kishcha et al., (2007) and Platnick et al., 33 (2015). 34 35 3. Retrieval of Cloud base height (CBH) and Lifting Condensation Level (LCL)

36

37 3.1 Cloud Statistics from the DL

1 We have used the vertical velocity and cloud statistics derived data of the DL during GVAX (Newsom et al., 2015). 2 In addition to clear-air vertical velocity statistics, we can derive the CBH, cloud fraction, cloud base vertical velocity 3 and cloud base updraft fraction. For the current dataset, the vertical velocity and cloud statistics value added 4 product (VAP) uses a 30-minute averaging time window, but produces output using a 10-minute sampling 5 interval. Thus, every third sample is statistically independent. The cloud fraction is the fraction of time during 6 the averaging interval that a cloud is detected at any height. Similarly, the cloud base updraft fraction is the fraction 7 of time that a positive (upward) cloud base vertical velocity is observed during the averaging interval. CBH 8 estimates are obtained by locating the heights of sharp spikes in the 1-sec range-corrected SNR profiles, as 9 illustrated in Figure 4. To minimize false detections, the CBH algorithm uses a method based on the first 10 derivative of the range-corrected SNR. When a cloud is present in the profile, the first derivative, which is 11 computed using a simple central-difference approximation, shows a strong positive peak immediately below 12 and a strong negative peak immediately above the cloud base. We require the magnitude of these peaks to 13 exceed 0.1 km, and separation to be between 2 and 15 range bins. If these conditions are satisfied, then the 14 algorithm locates the maximum in the range-corrected SNR between these two extrema. The height of this 15 maximum then determines the CBH. This process is then repeated for all 1-sec profiles acquired during a 16 given 24-hour period. Additional checks are then applied to minimize false detections by rejecting temporally 17 isolated CBH estimates. This is done by computing the absolute difference in CBH between a given profile 18 and the CBH values from profiles located immediately before and after in time. If both differences exceed 19 1km then that CBH value is rejected. Once the CBH values have been determined in this way, the cloud base 20 vertical velocity is determined from the vertical velocity at the CBH. The vertical velocity and cloud statistics 21 VAP reports the median value of the 1-sec CBH values and cloud base vertical velocities over a given 30-min 22 averaging interval. Further details are given in Newsom et al. (2015).

23

**24 3.2 CBH retrieval by using CM**

25

The measurement of the CBH with CM is known as standard method of the ground-based active remote sensing technique. The time delay between the transmitted and backscattered signal from the haze, fog, virga, mist and precipitation to the receiver of CM can be used to estimate the CBH. By knowing the time delay in equation (1), CBH can be estimated as **function of height with atmospheric visibility threshold**

- 30 Cloud base height (h) = (c\*t/2)
- 31

Cloud base height (h) = (c\*t/2) (1)

where c (= 3 x 108 m s-1) is the speed of light and t is the time delay. The backscattering coefficient is estimated by using the strength and attenuation of the backscattered signal from the atmosphere. Cloud base is identified by the strong increase of the backscatter coefficient and three layers of clouds can be detected if the lower clouds are transparent (Emeis et al., 2009; Morris, 2012). Flynn, (2004) have developed an algorithm to determine the cloud base as the height when they have observed a reduction in the visibility order of 100 m in the atmosphere. We have used the standard output of CBH from Väisälä laser Ceilometer (CT25K) and it uses the visibility threshold method for CBH (Väisälä Oyj, 2002). The CeiLinEx (Ceilometer Performance
 Experiment at Lindberg: http://ceilinex2015.de/special-topics/test.) 2015 showed that the retrieval of CBH
 leads to different results with the different algorithms.

**4 **3.3 CBH Retrieval by MODIS**

5

6 The estimation method of the CBH by using MODIS is described in detail by Hutchison (2002) and Sharma et 7 al., (2016). In order to estimate the CBH over the Manora Peak by using the MODIS Terra dataset, we have 8 used the cloud top pressure, cloud optical thickness, effective radius of the water cloud particle and liquid 9 water path during daytime from MODIS Terra satellite over the observational site for cloud passages 10 observed by the TSI. The CBH from the MODIS is calculated by taking the difference between the cloud top 11 height and the thickness of cloud ( $\Delta Z$ ) which is given in equation (2).

12
$$Z_{\text{Cloud base height}} = Z_{\text{Cloud top height}} - (\Delta Z)$$

(2)

(4)

13 where  $\Delta Z$  is the cloud thickness and  $\Delta Z$  is ratio of LWP and LWC.

1

The thickness of water cloud depends on the relation between liquid water path (LWP) and liquid water
 content (LWC). Liou (1992) showed that the relation of cloud optical thickness (τ) and effective radius of cloud

16 particle size  $(r_{eff})$  with LWP is given by

$$LWP = (2*\tau * r_{eff})/3 \text{ g.m}^{-2}$$
(3)

18 LWC=0.26 g.m-3 taken for cumulus cloud in clean condition (Hess et al., 1998).

 $\Delta Z = (LWP/LWC)$

20 By using LWP & LWC in equation (4), we have calculated the thickness of cloud ( $\Delta Z$ ).

21 We have estimated the CBH for water cloud present in the atmosphere by using equation (2) & (4).

22

24

17

23 3.4 Lifted condensation level estimation by using surface MET and RS datasets

- 25 The estimation of water vapor content from surface MET data has been derived by using equation (5) with T and
- 26 RH of surface meteorology (Goff-Gratch, 1946).

$$\mathbf{e}_{\mathrm{s}} = \mathbf{e}_{\mathrm{st}} * 10^{\mathrm{Z}} \tag{5}$$

28 where

29

30

27

$$Z = A\left(\frac{T_{S}}{T} - 1\right) + B \times \log_{10} \left(\frac{T_{s}}{T}\right) - C \times \left[10^{D\left(1 - \frac{T}{T_{S}}\right)} - 1\right] + F\left[10^{H\left(\frac{T_{S}}{T} - 1\right)} - 1\right]$$

and A = -7.90298, B= 5.02808, C=-1.3816 X  $10^{-7}$ , D= 11.344, F=8.1328 X  $10^{-3}$ , H= -3.49149 are the constants. est (=1013.246 mb) is saturation vapor pressure (es) at boiling temperature (Ts=373.16 K) at standard atmospheric pressure. By using saturation vapor pressure (es) from equation (5) and surface RH in equation (6), we have calculated the water vapor pressure (e)

$$\mathrm{RH} = \left(\frac{\mathrm{e}}{\mathrm{e}_{\mathrm{g}}}\right) \times 100 \tag{6}$$

2 Dew point temperature  $(T_d)$  estimation by using surface MET vapor pressure (e) is given by the equation (7) and it

3 is taken from the Lawrence, (2005)

$$\boldsymbol{T}_{d} = \begin{pmatrix} \mathbf{1} \\ \overline{\left[\left(\frac{1}{T_{0}}\right) - \left(\frac{R_{v}}{L}\right) * ln\left(\frac{e}{e_{0}}\right)\right]} \end{pmatrix}$$
(7)
kPa,  $\frac{R_{v}}{L} = 0.0001844 \text{ K}^{-1}$ , e - vapor pressure

5 where  $T_0=273$  K,  $e_o = 0.611$  kPa,

6 By knowing the temperature (T) and dew point temperature ( $T_d$ ) from surface meteorology and RS, we have 7 calculated LCL by using equation (8)

8

Lifting Condensation level (LCL) height (km) =  $0.125^*$  (T-Td) (8)

9

**15 4. Results and discussion**

16

17 Figure 1 shows one of the six (12 October 2011, 21 November 2011, 11 December 2011, 20 January 2012, 08 18 February 2012 and 14 March 2012) cloud case examples considered in this study observed by TSI for the estimation 19 and comparison of CBH by different instruments over the observational site. It shows the raw (Figure 1a) and 20 masked (Figure 1b) cloud images by TSI at hourly interval during daytime from (10.5-15.5 hr) on 12 October 2011. 21 The "yellow dot" in the TSI masked image represents the position of the sun, not obscured by the clouds. However, 22 if this "yellow dot" becomes "white" then the sun is obscured completely by the clouds (Figure 1b; Pfister et al., 23 2003). There is a difference between the raw and masked images. In masked images, TSI software masks out 24 obstructions-the imager, its arm and the sun-blocking band in the raw images. It is also to be noted that the 25 presence of cloud is clearly apparent with the raw image of the sky captured by TSI (Figure 1a). However, the 26 masked images strongly confirm the presence of clouds and further distinguish between the thin and opaque clouds 27 by the color of the image. For instance, the blue, gray, and white colors in Figure 1b represent the cloud free-sky, 28 thin and opaque clouds, respectively. While the black color in Figure 1b represent the masked pixels which are not 29 used in determining the macrophysical property of cloud by the TSI. Temporal variation of masked images of clouds 30 captured by TSI for all cases in the 160° field of view (FOV) centered at zenith in the cloud images during 10.5-15.5 31 LT is shown in Figure 2. Due to masked sky images, the loss of about 17 % of the hemispherical solid angle of the 32 sky dome is resulted. In the analysis of clear/cloudy pixels, these masked 'black' parts are ignored (Long et al., 33 2006). From Figure 2, it is clearly seen that there are lesser clouds in the forenoon (before 12.5 LT) in comparison

to afternoon (after 12.5 LT) on 12 October, 21 November and 11 December 2011. We have observed the clouds at every hour on 20 January, 08 February and 14 March 2012. In Figure 3 (a-f), we also show the temporal variation of the percentage occurrence of thin (shown by black line with black open circle) and opaque clouds (red line with red open rectangle) for all the six cloud overpasses over the observational site. In most of the cases of figure 3(a-f), the percentage of opaque clouds are greater than percentage of thin clouds. The dominance of opaque and thin clouds is clearly seen from the figure 3(a-f) during daytime over the site. It is also evident from Figure 3 that the percentage occurrence of opaque clouds is more frequent over the site relative to the thin clouds during the

8 observational period.

9 Figures 5 (a1-f1) and (a2-f2), illustrate the height-time variation of SNR and backscatter for different cases 10 of cloud passage over the observational site observed by DL and CM, respectively. Figure 5 depicts the presence of 11 ABL clouds over the site. The development of convective clouds in the lowest part of ABL is due to the presence of 12 convective thermals. These convective thermals are crucial in the formation of the clouds because these thermals can 13 rise from the surface to the top of the mixing layer without being diluted (Crum and Stull, 1987). It should be noted 14 that the presence of the convective clouds in the ABL can be confirmed by using the observed CBH from DL and 15 CM and lifted condensation level (LCL) estimated from the surface (Stull and Eloranta, 1985; Zhang and Klein, 16 2013). During the convection, the maximum SNR is observed due to the presence of low level ABL cumulus clouds. 17 Also, the observed cloud cases show different dynamics of the cumulus clouds over the site. Figure 5(a1) shows the 18 SNR maximum around 11.5-12.5 LT showing high percentage of opaque cloud during 12.5-14.5 LT and then 19 dominated by a thin clouds, consistent with Figures 5(a1) and 5(b1). Other cases also depict similar variation with 20 opaque clouds more frequent than the thin clouds during convection (see Figures 5b1-f1). Similarly, Figure 5 (a2-f2) 21 shows the height-time variation of averaged backscatter (srad-1.km-1.10-4) by the CM observed for all cloud cases in 22 the study. It is interesting to note that the temporal evolution and duration of thin and opaque clouds observed by 23 the TSI are in reasonable agreement with the DL and CM cloud pattern during all events.

24 In Figure 6 (a-f), we have plotted the temporal variation of CBH (with DL & CM) and cloud occurrence 25 frequency (with DL). The detailed description about the estimation of CBH is given in the section 3.1. The 26 fraction of time that a cloud is detected at any altitude during the given averaging period is defined as cloud 27 frequency. Varikoden et al. (2011) showed that the occurrence of low level clouds are more in comparison to the 28 mid-level clouds by using CM over a tropical station Akkulam, Thiruvananthapuram (8.29° N, 76.59° E, 15 m 29 above sea level) in India. They have also showed that the occurrence of low level clouds is higher during the 30 afternoon hours. We have also found that the frequency of occurrence of clouds are showing different 31 characteristics during forenoon and afternoon in the observed cases with both CM and DL over a high altitude 32 site.

Figure 7 (a-f) depicts the temporal variation of CBH observed by the DL and CM along with lifted condensation level (LCL) height estimated by using surface MET parameter and RS for all the selected case examples in the current study. There is a strong co-relation between the CBH observed by the DL and CM for all cases. On an average, the CBH from both the instruments is higher during the convective period and is associated with the change in LCL in the ABL during daytime. We have estimated the LCL with surface MET and RS to

1 compare with the CBH of DL and CM. In 12 October 2011 case, a small difference is observed between the CBH 2 (DL) and LCL heights but LCL heights with the MET and RS shows a similar pattern as CBH (CM) implying the 3 strong association with ABL dynamics (Jones et al., 2011). From Figure 7 (a-f), it is clearly observed that in all the 4 cases, CBH is coupled with the LCL estimated from the surface meteorological parameters. This strong dependence 5 of CBH with LCL suggests the link between cloud formation and development of convection on the surface (Zheng 6 et al., 2015; Zheng and Rosenfeld, 2015). From Figure 7, it is clearly observed that an overestimation 7 (difference observed between DL and CM CBH~ 0.5 km) of CBH is done by the DL in comparison to CM. 8 This could be due to different technical specification and retrieval techniques of both instruments. Similarly, 9 we have also observed the difference between LCL and derived CBH because of their retrieval techniques.

10 In Figure 8, we have plotted the temporal variation of CBH with cloud base vertical velocity for all cases. 11 CBH observed with both the instruments are showing similar temporal variation throughout the observational time 12 period (10.5-15.5 LT). From figure 8(a-f), it is clearly evident that the updrafts are dominant due to the diurnal 13 evolution of convective ABL during daytime over the site. The observed diurnal pattern of the vertical velocity 14 with DL for all the cases are showing the dominance of updrafts over the site. In some cases like 12 October 15 2011, 21 November 2011 and 08 February 2012, the vertical velocity follows the similar pattern. We have also 16 plotted the temporal variation of cloud base vertical velocity with cloud base vertical velocity updraft fraction (m) 17 for all cases in Figure 9 (a-f). From this figure, it is clearly seen that both the parameter are well correlated. We have 18 also compared the CBH calculated by the DL and CM with the MODIS derived CBH for all cloud passes over the 19 observational site. For instance, Figure 10 shows the MODIS Terra derived CBH and the daily mean (05-10 UT) 20 CBH measured by the DL and CM. We have taken the mean of latitude/longitude  $\pm 1$  degree by centering the 21 latitude/longitude of the observational site. The observed CBH from MODIS is well within the estimated standard 22 deviation from ground based CBH. It shows reasonably good agreement with the estimation of CBH from the 23 ground based and DL and CM CBH in all the cases except in two cases (21 November 2011 and 14 March 2012) 24 where the differences are slightly higher and need to be investigated for the possible inconsistencies. In Figure 10, 25 we have observed an overestimation of the MODIS CBH with respect to error bars of the observed CBH 26 from DL and CM. This overestimation of CBH by the MODIS could be due to the overpass and large spatial 27 grid.

28 Further, we have used the DL and CM CBH as well as cloud updraft and cloud base vertical velocity 29 observed by the DL for all six cases to see the correlation which is plotted in Figure 11. The correlation of CBH 30 between the DL and CM is shown in Figure 11a. It is noticed that the CBH estimated by the DL is well correlated 31 (R2=0.81) with the CM measured CBH when we combine all the cases shown in Figure 11a. We have observed 32 differences between the CBH DL and CM ~ 500 m which represents an overestimation of CBH by DL in 33 comparison to CM. These differences could be due to different methodologies for the estimation of CBH with both 34 the instruments and to the complex topography itself. ABL also plays an important role in the formation of clouds 35 and all the cases are during convective boundary layer conditions and this may be one of the reason behind the 36 overestimation with the DL in some cases. In addition, Figure 11b illustrates the relation between cloud base vertical 37 velocity and cloud updraft fraction observed by DL for all cloud passes over the observational site. As indicated in

1 Figure 11b, a strong correlation ( $R^2=0.71$ ) is also noted between these two parameters. Further, it is noticed that 2 when the cloud updraft fraction is less than 40%, the cloud base vertical velocity tends to be negative. However, 3 positive vertical velocities are noted when the cloud updraft fraction is more than 50%. Kollias et al. (2001) showed 4 that the cloud base vertical velocity is consistent with the updraft speed. We have also observed similar behavior 5 between the cloud base vertical velocity and updraft fraction although our observations are from a high altitude 6 location. Jeong and Li (2010) estimated the CBH by using micropulse Lidar for few case studies by applying the 7 threshold condition of aerosol particle diameter less than 1 µm and relative humidity 40 % over the southern great 8 plain site. They have observed the cumulus cloud on all cases and found the CBH varying in between 1-4 km, above 9 mean sea level (amsl). A detailed comparison of CBH estimated over various parts of the world by using different 10 ground based instruments and satellite datasets is shown in Table-2. Despite different site morphologies, our CBH 11 values observed with both DL and CM (Table-2) are in agreement with past studies across the globe. Bühl et al., 12 (2015) observed the cloud and vertical velocity by using different ground based instruments e.g. DL, cloud radar and 13 wind profiler over meteorological observatory, Lindenberg, Germany.

14 The cloud observations with DL & CM during all case examples show that CBH varies between ~ 2-3 km, 15 above mean sea level (amsl) over the site. The presence of higher magnitude (high positive vertical velocity) 16 updrafts in the cloud layers was also observed. The observed vertical velocity in the cloud layer varied between  $\pm$ 1.5 ms-1. Similar characteristics were observed at Manora Peak, Nainital. We have observed that cloud base vertical 17 18 velocity varies between  $\pm 2 \text{ ms}^{-1}$  except for 20 January 2012 during which higher vertical velocities of 0-4 ms-1 were 19 obtained. The observed CBH also varies between 2.3-2.7 km amsl in both instruments over Manora Peak, Nainital. 20 Hirsch et al. (2011) retrieved the CBH by CM, and observed the shallow cumulus cloud during daytime and CBH at 21 1.6±0.3 km, amsl. Also, Meerkötter and Bugliaro, (2009) estimated the CBH by using MSG/SEVIRI, NOAA 22 satellite data and CM data for convective cloud cases over the seven test stations near Germany and neighboring 23 countries. By using geostationary satellite and ground based CMs, they have observed that CBH varies between ~ 24 2-3 km and also showed a significant correlation. Thus, our results are in good agreement with the temporal 25 variation of CBHs observed by DL compared with CM in previous studies. The cooling and warming of the 26 atmosphere is governed by the presence of clouds at different altitudes in the atmosphere (Kiehl and 27 Trenberth, 1997). CBH of low level clouds coupled with shallow convection is playing an essential role in the 28 parameterization of weather and climate models (Chandra et al., 2015). Also the uncertainty observed in 29 climate models is due to low-level clouds (Bony and Dufresne, 2005) especially when model grid spacing is 30 much larger than the size of low level. Therefore, the continuous estimation of CBH will be a useful input for 31 the models. Further, the cloud radiative cooling, relative humidity in the ABL and cloud cover have direct 32 association with the low altitude clouds (Brient and Bony, 2012). Therefore, the accurate and systematic 33 measurements of low level cloud base become important for the improvement of the models. Hence, in this 34 report we investigated the potential of the DL in measuring the CBH over the site in comparison to CM. 35 From the current study, it is also clearly seen that we can use DL for CBH study over the site. It also 36 demonstrates that the precise observations of the CBH over the complex topography are very useful for 37 model validation.

**2 5. Summary and Conclusions**

3 In this study, we have presented comparison of the CBH estimated by using the DL with CM and MODIS derived 4 CBH over a high altitude site in the foothills of the Himalayan mountain range region. TSI shows the presence of 5 cloud over the site for the cases evaluated in the current study and also opaque clouds are more frequently observed 6 than thin clouds over the site during the observational period. The height-time variation of SNR of DL and 7 backscatter by the CM depict a similar pattern for the cases evaluated with opaque (thin) clouds dominating during 8 morning (afternoon) hours in most of the cases. Strong correlation (R2=0.81) between DL and CM CBH is observed 9 suggesting that DL can also be used as a potential instrument for measuring CBH apart from standard instrument 10 CM. Similarly, we have observed the good correlation ( $R^2$ =0.71) between cloud base vertical velocity and cloud 11 updraft fraction. We have observed a similar temporal variation between CBH (estimated from DL and CM) and 12 LCL height (Surface MET and RS) during all the cases. The CBH height and LCL height derived from surface MET 13 and RS are also comparable. The estimated CBH with the MODIS data is also in close agreement with the ground 14 based instruments in most of the observed cases. 15 Further, our results also show close agreement with the CBH derived by DL, CM and MODIS derived satellite

data sets in all cases. We have also noticed an overestimation (~ 500 m) of CBH by DL in comparison to CM which is due to different technical specifications of the instruments and different retrieval methodologies of CBH. Similarly, a difference is observed between the MODIS derived CBH and DL, CM derived CBH mainly due to the large spatial grid and overpass time of the MODIS over the observational site. By considering the importance of the current study, CBH estimations by DL along with the cloud updraft velocities will be utilized in our future studies as potential inputs for numerical weather prediction models over the foothills of the Himalayan mountain range region.

- 23
- 24
- 25

**26 Acknowledgments**

27

This work has been carried out as a part of GVAX campaign under joint collaboration among Atmospheric Radiation Measurement (ARM), Department of Energy (US), Indian Institute of Science (IISc) and Indian Space Research Organization (ISRO), India. We thank Director, ARIES for providing the necessary support. We also thank MODIS team for providing the valuable datasets used in the present study. The authors wish also to thank the Associate Editor, for his judicious comments which helped improve the quality of the paper. **The authors thank to Victor Morris for his detailed discussion regarding the Total Sky Imager (TSI) and his inputs to improve the**

- 34 quality of work.
- 35

Table-1 Technical specification of the Doppler Lidar operated over the Manora Peak during GVAX

Manufacturer **Halo Photonics** Eye safety Class 1M Wavelength 1.5 μm Laser pulse energy ~100 µJ 200 ns Laser pulse width Pulse rate 15 kHz 19.4 ms-1 Nyquist Velocity Unambiguous range 10 km 75 mm Aperture 0.5 m3 Volume approximately Power consumption < 300 W Mass approximately 85 Kg Temporal resolution selectable from 0.1 to 30 seconds 18 to 60m Range gate size  $< 20 \text{ cm s}^{-1} \text{ for SNR} > -17 \text{ dB}$ Velocity precision Minimum range <100m, typically 75m

3

Scanning

Enclosure

- 4 5
- 6

7

8

9

5

10

Step-stare, full upper hemisphere

Weatherproof, temperature stabilized

1 Table-2: Comparison of cloud base heights with other locations around the world

| Observational site
(Latitude/longitude/elev
ation)              | Instrument                                      | Date                                                              | Cloud base height
(km)
(amsl)                                                                                                                 | References                       |
|-----------------------------------------------------------------------|-------------------------------------------------|-------------------------------------------------------------------|-----------------------------------------------------------------------------------------------------------------------------------------------------|----------------------------------|
| Lindenberg, Germany.                                                  | Doppler Lidar,
Cloud radar,
wind profiler | 30 July 2013                                                      | 2.9 km                                                                                                                                              | Bühl et al.,2015                 |
| Israel
(31.89 0 N , 34.81 0 E, 60
m)      | Ceilometer                                      | 22 April 2010                                                     | 1.6±0.3 km                                                                                                                                          | Hirsch et al., 2011              |
| Southern Great Plain
(36.6 0 N , 97.5 0 W)   | Micro pulse
Lidar                            | 07, 13 and 22 May 2003                                            | 4.2, 1.6 and 1.3 km, respectively                                                                                                                   | Jeong and Li,2010                |
| Seven test station near
Germany and neighboring
countries       | MSG/SEVIRI
NOAA
Ceilometer                | 23,30 May and 30
July 2007                                     | Between 2-3 km                                                                                                                                      | Meerkötter and
Bugliaro, 2009 |
| Nainital, India
(29.4 0 N, 79.2 0 E, 1958 m) | Ceilometer
Doppler Lidar                     | 12 Oct, 21 Nov,
11 Dec, 2011
20 Jan, 08 Feb,
14 Mar 2012 | 2.468       2.328         2.298       2.228         2.688       2.568         2.438       2.418         2.678       2.658         2.348       2.258 | Current study                    |

- \* Red color in table represents the CBH of Ceilometer